# FreqKV: Frequency Domain Key-Value Compression for Efficient Context Window Extension

## Abstract

Extending the context window in large language models (LLMs) is essential for applications involving long-form content generation. However, the quadratic complexity of self-attention and the linear increase in key-value (KV) cache memory requirements with respect to sequence length present significant challenges during fine-tuning and inference. Although LongLoRA achieves efficient fine-tuning by employing shifted sparse attention, inference remains inefficient due to the requirement for dense global attention. In this work, we introduce a novel context extension method that optimizes both fine-tuning and inference efficiency. Our method exploits a key observation: in the frequency domain, the energy distribution of the KV cache is primarily concentrated in low-frequency components. By filtering out the high-frequency components, the KV cache can be effectively compressed with minimal information loss. Building on this insight, we propose an efficient compression technique, FreqKV, that iteratively reduces the increasing KV cache to a fixed size in the frequency domain, applicable to both fine-tuning and inference. With minimal fine-tuning, LLMs can learn to leverage the limited cache that is compressed in the frequency domain and extend the context window efficiently. FreqKV introduces no additional parameters or architectural modifications, ensuring compatibility with the original full attention post-training. Experiments on long context language modeling and understanding demonstrate the efficiency and efficacy of the proposed method.

## 1 Introduction

Large language models (LLMs) typically have a limited size of context window, which is pre-defined during the pre-training process. However, it is inevitable for LLMs to process sequences that exceed the preset context size. LLMs struggle to maintain their performance when generalized to longer contexts. Additionally, the computation cost of the self-attention mechanism (Vaswani et al., 2023) grows quadratically with the context length, meaning that doubling the context window results in a fourfold increase in the computational cost of attention modules.

For efficiency, existing efforts aim to compress the key-value (KV) cache for long contexts during inference. They evict (Xiao et al., 2024; Li et al., 2024) or merge (Zhang et al., 2024b; Wan et al., 2024) KV states of less important tokens following certain rules. They use attention scores to measure the importance and approximate the original full attention. However, while these methods provide an approximation of full computation on existing tokens through different strategies, they can not fully prevent performance degradation when decoding future tokens.

Recent studies propose to fine-tune LLMs to longer contexts to extend the context window. LongLoRA (Chen et al., 2024) trains LLMs using shifted sparse attention. Despite training efficiency, their sparse attention fails to be applied during inference, and they still require the original attention on the full sequence. Concurrently, LoCoCo (Cai et al., 2024a) and Activation Beacon (Zhang et al., 2024a) introduce additional modules to compress KV states. They incorporate the fine-tuned compressing pattern into the decoding procedure of LLMs.

In the field of computer vision, studies have shown that low-frequency channels are more important for convolutional neural networks (CNNs) (Xu et al., 2020). Moreover, Fourier Transformer (He

et al., 2023) discards the high-frequency parts of the contexts and downsample the hidden states in the encoders. Inspired by these work, we seek to compress KV states in the frequency domain without the need for additional compression modules in decoder-only LLMs. We transform key states and value states in LLaMA-2-7b (Touvron et al., 2023) from the time domain to the frequency domain for power spectrum analysis. As shown in Figure 1, the energy distribution increasingly concentrates on low-frequency components as the computation process progresses inside the model. This suggests that high-frequency components, which contribute less to the overall information, can be discarded without a significant impact on performance, thereby enhancing computational efficiency.

In this paper, we introduce FreqKV, an efficient context extension method that iteratively compresses key-value states in the frequency domain. Compression is triggered only when the KV cache reaches the predefined context window size. We keep the first few tokens uncompressed due to the attention sinks of LLMs (Xiao et al., 2024; Han et al., 2024). During each compression step, the low-frequency components of the KV states are preserved at a specified retaining ratio. Subsequent tokens are appended to the compressed cache until it is filled again. This ensures that the maximum number of cached KV states that each query token can attend to is limited below the context window size. To reduce memory and computational costs, the compressed cache will be further compressed together with the incoming tokens. This iterative compression mechanism leads to an increased compression level of the earlier contexts as the sequence length grows. Without introducing additional compression modules, LLMs could learn to utilize the compressed cache efficiently when extending to longer contexts. FreqKV demonstrates comparable performance to other methods that employ full KV cache or additional compressors in long context language modeling. Furthermore, experiments on LongBench (Bai et al., 2024) indicate that FreqKV surpasses recently studied KV compression methods in long-context understanding, achieving higher scores on open-ended text generation tasks.

## 2 RELATED WORK

**KV Compression for LLMs.** To extend the context window of LLMs efficiently, researchers attempt to compress the KV cache as more tokens are fed into the model. One common approach is selective token eviction (Xiao et al., 2024; Li et al., 2024; Cai. et al., 2024b), where less significant tokens are discarded. Although the eviction strategies ensure that the size of KV cache involved in each decoding step does not exceed the pre-defined context window size, LLMs suffer from the permanent loss of the information associated with evicted tokens. To address this limitation, some researchers introduce cache merging techniques to approximate the original full attention of the existing contexts (Zhang et al., 2024b; Wan et al., 2024; Wang et al., 2024). However, these inference methods often sacrifice performance for efficiency.

**Context Extension for LLMs.** Recent advancements in context extension for LLMs have focused on efficiently scaling models to handle longer input sequences without significantly increasing computational costs. LongLoRA (Chen et al., 2024) employs shifted sparse attention during the parameter-efficient fine-tuning. However, this sparse attention mechanism is not applicable during inference, necessitating a return to the original full attention post-training. Other techniques, such as LoCoCo (Cai et al., 2024a), integrate convolutional operations into LLMs for compressing long contexts. They fine-tune the compression modules together with LLMs. Landmark attention (Mohtashami & Jaggi, 2023) uses landmark tokens to retrieve previous input blocks. Similarly, Activation Beacon (Zhang et al., 2024a) proposes the use of a special token to represent the previous context for compression. However, they introduce a copy of multi-head attention parameters, which can amount to approximately 2 billion for 7 billion parameter models. In contrast, our proposed method achieves context extension without introducing any additional parameters.

**Learning in the Frequency Domain.** Learning in the frequency domain is a well-established technique to compress images and accelerate CNNs (Gueguen et al., 2018). It has been observed that CNNs are more sensitive to low-frequency channels than high-frequency channels (Xu et al., 2020). These works have inspired efforts to process natural language. FNet (Lee-Thorp et al., 2022) enhances the efficiency of Transformer encoder architectures by replacing the self-attention layers with the Fourier transform to serve the purpose of mixing tokens. Additionally, Fourier Transformer

(He et al., 2023) eliminates redundancies in the context through frequency domain processing within encoder architectures.

However, because of the auto-regressive nature, it remains unclear how to leverage frequency components for decoder-only Transformer, which is the main architecture of generative LLMs. To the best of our knowledge, FreqKV is the first work that explores compressing key-value states in the frequency domain for decoder-only LLMs.

## 3 PRELIMINARIES

### 3.1 DISCRETE COSINE TRANSFORM

The Discrete Cosine Transform (DCT) transforms a signal from the spatial domain (time or position) into the frequency domain. Several variants of the DCT exist, with DCT-II being the most common. For a real-value discrete signal $\boldsymbol{X}_{0:N-1} = [x_0, \ldots, x_{N-1}]$ of length $N$, it is defined as:

$$y_t = \alpha_t \sum_{n=0}^{N-1} x_n \cdot \cos\left[\frac{\pi t(2n+1)}{2N}\right], \quad \alpha_t = \begin{cases} \sqrt{\frac{1}{N}} & \text{if } t = 0, \\ \sqrt{\frac{2}{N}} & \text{otherwise} \end{cases} \quad (1)$$

where $t = 0, 1, \cdots, N-1$. $\alpha_t$ is the normalization factor. The original signal $\boldsymbol{X}_{0:N-1}$ can be recovered by applying the inverse DCT (IDCT) on the frequency components $\boldsymbol{Y}_{0:N-1}$:

$$x_n = \sum_{t=0}^{N-1} \alpha_t \cdot y_t \cdot \cos\left[\frac{\pi t(2n+1)}{2N}\right]. \quad (2)$$

The frequency components are expressed as a combination of the original signals. The values can be computed using the Fast Fourier Transform (FFT) with a complexity of $O(N\log N)$. The amplitudes of frequency components are utilized in the power spectrum analysis to represent the energy or magnitude of components. The components of higher energy in the frequency domain indicate they are more informative (He et al., 2023).

### 3.2 SELF-ATTENTION

For the incoming token $x_N$, the prefilled $N$ tokens $X_{0:N-1}$ are utilized as the cache during decoding. Denote the hidden states of the $N+1$ tokens input to a specific layer of LLMs as $\boldsymbol{H}_{0:N} = [\boldsymbol{h}_0, \cdots, \boldsymbol{h}_N]$. The query, key and value states of $\boldsymbol{x}_N$ are are computed as follows::

$$\boldsymbol{q}_N = \boldsymbol{h}_N \boldsymbol{W}^Q, \quad \boldsymbol{k}_N = \boldsymbol{h}_N \boldsymbol{W}^K, \quad \boldsymbol{v}_N = \boldsymbol{h}_N \boldsymbol{W}^V, \quad (3)$$

where $\boldsymbol{W}^Q, \boldsymbol{W}^K, \boldsymbol{W}^V$ are the projection matrices for the query, key and value states, respectively. For simplicity, indices corresponding to layers and heads have been omitted.

The cached KV states for the previous $N$ tokens $\boldsymbol{X}_{0:N-1}$ are:

$$\boldsymbol{K}_{0:N-1} = \boldsymbol{H}_{0:N-1} \boldsymbol{W}^K, \quad \boldsymbol{V}_{0:N-1} = \boldsymbol{H}_{0:N-1} \boldsymbol{W}^V. \quad (4)$$

When calculating attention scores, the incoming token $x_N$ attends to all cached KV states as well as to itself:

$$\mathcal{A}(N) = \text{Softmax}\left(\frac{\boldsymbol{q}_N[\boldsymbol{K}_{0:N-1} \oplus \boldsymbol{k}_N]^T}{\sqrt{d}}\right) \cdot [\boldsymbol{V}_{0:N-1} \oplus \boldsymbol{v}_N], \quad (5)$$

where $d$ is the hidden dimension. $\oplus$ means the concatenation of the KV cache and KV states of $x_N$

## 4 FREQKV

### 4.1 ENERGY CONCENTRATION IN THE FREQUENCY DOMAIN

We transform key states and value states from the time domain, which is the sequence dimension, to the frequency domain. The average power spectrums in different decoder layers of LLaMA-2-7b are calculated and presented in Figure 1.

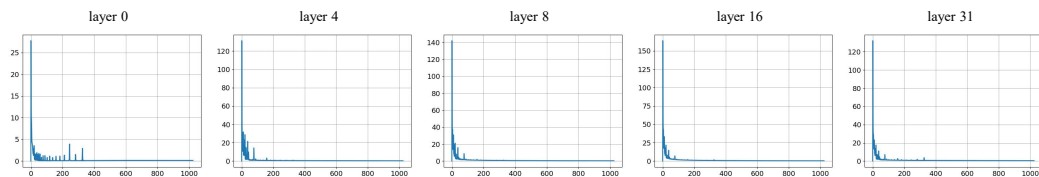

(a) The average power spectrums of key states.

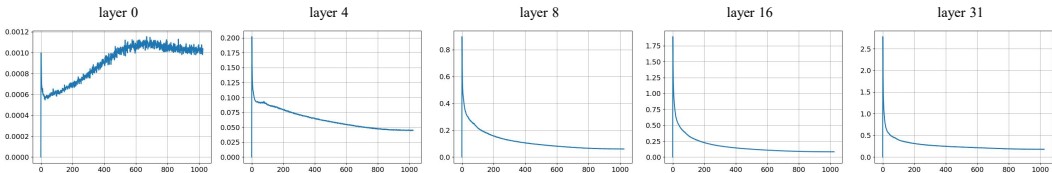

(b) The average power spectrums of value states.

Figure 1: The average power spectrums of key states and value states in different layers of LLaMA-2-7b. 1000 documents are sampled from CNN/Daily Mail (Hermann et al., 2015). We use DCT to transform key states and value states to the frequency domain, and average power spectrums over these samples and hidden dimensions.

The figure shows that the energy of key states and value states is increasingly concentrated in the low-frequency components. Although there is no obvious energy concentration in the frequency domain for the initial embeddings of natural languages such as value states from layer 0, the model tends to aggregate energy in the low-frequency components along the decoding procedure. The observation of energy concentration suggests that we could maintain low-frequency components and filter out high-frequency components which could be redundant. Head-wise analysis of the power spectrum is provided in Appendix A.

## 4.2 KV Compression in the Frequency Domain

We conduct DCT along the sequence dimension to transfer the KV cache to the frequency domain:

$$\boldsymbol{Y}^K_{0:N-1} = \text{DCT}(\boldsymbol{K}_{0:N-1}), \quad \boldsymbol{Y}^V_{0:N-1} = \text{DCT}(\boldsymbol{V}_{0:N-1}). \tag{6}$$

As observed in Figure 1, since the lower-frequency components are of higher magnitude and carry more information, we will retain them and remove higher-frequency components for compression. Given the retaining ratio $\gamma$, the retaining size is $L = \gamma \cdot N$. $N - L$ high-frequency components are filtered out to reduce redundancy:

$$\widetilde{\boldsymbol{Y}}^K_{0:L-1} = \boldsymbol{Y}^K_{0:N-1}[0:L-1], \quad \widetilde{\boldsymbol{Y}}^V_{0:L-1} = \boldsymbol{Y}^V_{0:N-1}[0:L-1]. \tag{7}$$

Then we conduct IDCT along the frequency dimension to convert the compressed components back to the time dimension. It should be noted that the time-domain signals are normalized by the square root of the component number as shown in the formula of IDCT (Equation 2). Therefore, the compressed signals should be rescaled with $\sqrt{\frac{L}{N}}$ to restore the original amplitude:

$$\widetilde{\boldsymbol{K}}^{0:N-1}_{0:L-1} = \sqrt{\frac{L}{N}}\text{IDCT}(\widetilde{\boldsymbol{Y}}^K_{0:L-1}), \quad \widetilde{\boldsymbol{V}}^{0:N-1}_{0:L-1} = \sqrt{\frac{L}{N}}\text{IDCT}(\widetilde{\boldsymbol{Y}}^V_{0:L-1}). \tag{8}$$

$\widetilde{\boldsymbol{K}}^{0:N-1}_{0:L-1}$ and $\widetilde{\boldsymbol{V}}^{0:N-1}_{0:L-1}$ are the KV cache of size L in the time domain. The superscript "$0:N-1$" means that $\widetilde{\boldsymbol{K}}^{0:N-1}_{0:L-1}$ and $\widetilde{\boldsymbol{V}}^{0:N-1}_{0:L-1}$ are the compressed KV of the cached $N$ tokens. The subscript "$0:L-1$" means the retaining size is $L$. The incoming token $x_N$ will attend to the compressed KV cache:

$$\widetilde{\mathcal{A}}(N, L) = \text{Softmax}\left(\frac{\boldsymbol{q}_N[\widetilde{\boldsymbol{K}}^{0:N-1}_{0:L-1} \oplus \boldsymbol{k}_N]^T}{\sqrt{d}}\right) \cdot [\widetilde{\boldsymbol{V}}^{0:N-1}_{0:L-1} \oplus \boldsymbol{v}_N]. \tag{9}$$

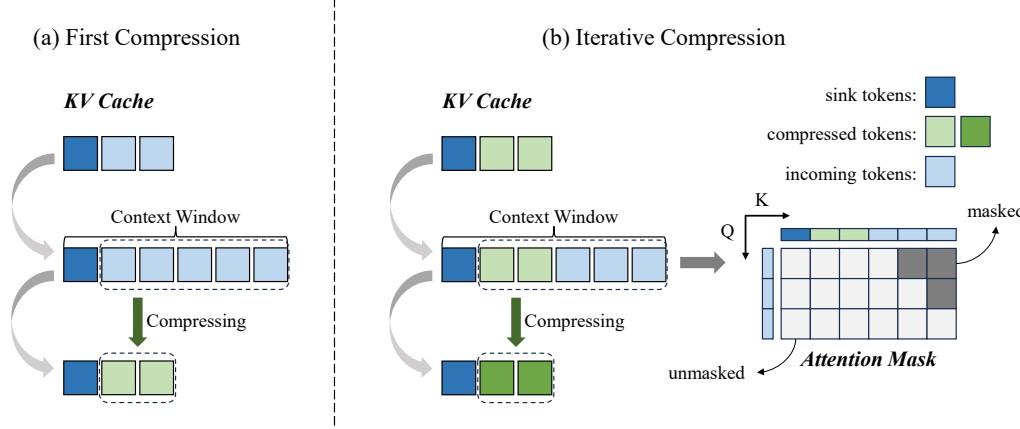

Figure 2: The illustration of our FreqKV. The KV cache will be compressed in an iterative manner as the cache reaches the context window size. Sink tokens remain uncompressed throughout the process. (a) The tokens after sink tokens will be compressed in the frequency domain and subsequent tokens will continue to get into the cache. (b) When the cache is filled again, the compressed tokens and incoming tokens will be compressed together. The compression is performed iteratively to extend the context window.

### 4.3 CONTEXT EXTENSION WITH THE COMPRESSED KV

When extending the context window, the memory requirement of KV cache increases linearly and the computation cost of the original full attention grows quadratically with the length. To limit the size of KV cache for each query token to attend below the context window size, we will compress KV cache in the frequency domain iteratively as the cache is filled. We illustrate our FreqKV in Figure 2.

For tokens within the context window, the standard attention will be conducted. For tokens out of the window, we will compress the cached KV states in the frequency domain. Because the retaining size $L$ is smaller than the cache size $N$, subsequent tokens could get in and fill the cache. We discard the original KV states of the prefilling tokens and maintain the compressed KV states. They will be concatenated with the KV states of the incoming tokens and compressed together when the cache is filled again. Tokens that appear earlier in the sequence undergo more iterations of compression as the context window expands, whereas less compression will be performed on the more recent tokens. With the iterative FreqKV, the KV cache size is not fixed and is reduced below the context window size during decoding. Since the compression is only performed when the sum of cached KV and incoming tokens reaches the preset size, the computation overhead of the compression could be negligible. For example, with the context size of 4096 and the retaining size of 2048, compression is performed every 2048 tokens for contexts exceeding the original window.

Recent work has found the phenomenon of attention sinks that LLMs tend to assign high attention scores to initial tokens (Xiao et al., 2024; Han et al., 2024). Therefore, we maintain these initial tokens uncompressed in the cache and only compress tokens that come after them.

During training and the prefilling stage of inference, the whole sentence is tokenized and fed into the model. The attention is computed chunk-wise interleaved with the compression operation. After each compression, $N - L - S$ incoming tokens are regarded as a chunk and fill the cache, with $S$ sink tokens uncompressed. As shown in Figure 2 (b), the incoming tokens in each chunk can not attend to the subsequent tokens. The newly incoming token $x_M$ will attend to $S$ sink tokens, $L$ compressed "tokens", $M - N$ previous incoming tokens, and $x_M$ itself. The calculation of attention in Equation 9 can be reformulated as follows:

$$\widetilde{\mathcal{A}}(S, N, L, M) = \text{Softmax}\left(\frac{q_M[\boldsymbol{K}_{0:S-1} \oplus \widetilde{\boldsymbol{K}}_{0:L-1}^{S:N-1} \oplus \boldsymbol{K}_{N:M}]^T}{\sqrt{d}}\right) \cdot [\boldsymbol{V}_{0:S-1} \oplus \widetilde{\boldsymbol{V}}_{0:L-1}^{S:N-1} \oplus \boldsymbol{V}_{N:M}]. \quad (10)$$

Table 1: Perplexity evaluation on the test sets of PG-19 and Proof-pile. The superscript "*" means that we reproduce LoCoCo following their official code for evaluation. The results of full fine-tuning and LongLoRA are reported from Chen et al. (2024).

| Training Length | Method | Inference Cache | Evaluation Context Length | | | | |
|---|---|---|---|---|---|---|---|
| | | | 2048 | 4096 | 8192 | 16384 | 32768 |
| *PG-19* | | | | | | | |
| 8192 | Full FT | Full | 7.55 | 7.21 | 6.98 | - | - |
| | LongLoRA | Full | 7.70 | 7.35 | 7.14 | - | - |
| | LoCoCo* | Compressed | 8.15 | 8.08 | 7.27 | - | - |
| | FreqKV | Compressed | 7.53 | 7.19 | 7.13 | - | - |
| 16384 | LongLoRA | Full | 7.65 | 7.28 | 7.02 | 6.86 | - |
| | FreqKV | Compressed | 7.77 | 7.40 | 7.32 | 7.29 | - |
| 32768 | LongLoRA | Full | 8.29 | 7.83 | 7.54 | 7.35 | 7.22 |
| | FreqKV | Compressed | 8.14 | 7.73 | 7.61 | 7.56 | 7.54 |
| *Proof-pile* | | | | | | | |
| 8192 | Full FT | Full | 3.14 | 2.85 | 2.66 | - | - |
| | LongLoRA | Full | 3.20 | 2.91 | 2.72 | - | - |
| | LoCoCo* | Compressed | 3.40 | 3.20 | 2.88 | - | - |
| | FreqKV | Compressed | 3.16 | 2.88 | 2.80 | - | - |
| 16384 | LongLoRA | Full | 3.17 | 2.87 | 2.66 | 2.51 | - |
| | FreqKV | Compressed | 3.22 | 2.93 | 2.84 | 2.80 | - |
| 32768 | LongLoRA | Full | 3.35 | 3.01 | 2.78 | 2.61 | 2.50 |
| | FreqKV | Compressed | 3.34 | 3.03 | 2.93 | 2.88 | 2.86 |

## 5 EXPERIMENTS

### 5.1 IMPLEMENTATION

We conduct experiments on long context language modeling and understanding tasks with LLaMA-2-7b (Touvron et al., 2023) base and chat models. Minimal training is introduced to adapt the model to this frequency-domain compression method. For long context language modeling, we fine-tune LLaMA-2-7b on the RedPajama (Computer, 2023) pre-training dataset for 1000 steps, extending the context window size from 4K to 8K, 16K, and 32K. Perplexity (PPL) evaluation is conducted on PG-19 (Rae et al., 2019) and Proof-pile (Azerbayev et al., 2022). For long context understanding, the instruction following dataset LongAlpaca (Chen et al., 2024) is used for the supervised fine-tuning (SFT) of the chat model. The context window size is extended from 4K to 8K. The model is trained on 6.28K long-context QA samples for 5 epochs and evaluated on LongBench (Bai et al., 2024).

The total batch size ($GPU\_number \times Batch\_size\_per\_device \times Gradient\_accumulation\_steps$) is 64. The learning rate increases linearly from 1e-6 to 2e-5 with 20 warm-up steps and remains constant in the following steps. The rank used in the LoRA (Hu et al., 2021) fine-tuning is set to 8. Following LongLoRA (Chen et al., 2024), the embedding and normalization layers are learnable during training.

The preset context window size of LLaMA-2 is 4096, which is also the maximum KV cache size $N$. We maintain $S = 4$ sink tokens uncompressed. The retaining ratio $\gamma$ in compression is set to 0.5. Therefore, the retaining size during each compression is $L = \gamma \cdot (N - S) = 2046$. As long as the cache size reaches its capacity of 4096, the 4092 states since the 5-th state in the cache will be compressed into 2046 states. All the experiments are conducted on ADA6000 and RTX4090 GPUs. Moreover, we equip our method with FlashAttention-2 (Dao, 2023) for further acceleration and memory saving.

Table 2: Training time and memory usage of FreqKV when extending to 8K, 16K and 32K. All the statistics are collected with the same experimental settings.

| Training Length | Training Time (hours) | Memory Usage (GB) |
|---|---|---|
| 8K | 17.61 | 25.08 |
| 16K | 39.40 | 33.07 |
| 32K | 89.99 | 44.90 |

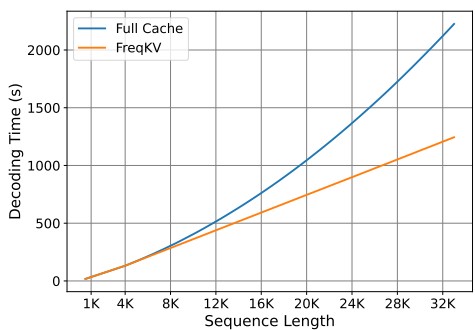

Figure 3: Decoding time with the full cache and FreqKV on the increasing sequence length.

## 5.2 LONG CONTEXT LANGUAGE MODELING

We use FreqKV to train the model on RedPajama (Computer, 2023) with lengths of 8192, 16384, and 32768. Perplexity is measured on test sets of the book corpus dataset PG-19 (Rae et al., 2019) and the Arxiv math dataset Proof-pile (Azerbayev et al., 2022) with the evaluation sliding window of 256. We compare our method with other baselines including full fine-tuning (Full FT), LongLoRA (Chen et al., 2024), and LoCoCo (Cai et al., 2024a). While Full FT and LongLoRA leverage full KV cache during inference, LoCoCo, and our FreqKV use compressed cache.

PPL scores on different evaluation context lengths are reported in Table 1. Although FreqKV employs an iterative compression manner, its performance does not deteriorate on extended context length. When compared to Full FT and LongLoRA, our compressed cache method effectively extends the context window without sacrificing much performance. Moreover, our method outperforms LoCoCo on the extended context length (8192) as well as the shorter lengths (2048 and 4096).

## 5.3 MEMORY AND COMPUTATIONAL COST

In Table 2, we present the training time and memory usage of FreqKV when extending to different context lengths. All statistics are collected under the condition that $Batch\_size\_per\_device = 1$ and $Gradient\_accumulation\_steps = 8$ with 8 ADA6000 GPUs.

With FreqKV, we can conduct training to extend the context window size of LLaMA2-7b from 4K to 32K. While 49GB memory is required by LongLoRA and 50GB for LoCoCo when extending to 16K (Cai et al., 2024a).

Furthermore, we compare the decoding time required by the full cache and our FreqKV when the sequence length increases. As shown in Figure 3, the decoding time starts to diverge at the length of 4K. While the full cache utilization leads to a quadratic growth in decoding time, the decoding time of FreqKV increases approximately linearly with a negligible time spent on compression, showcasing its efficiency.

## 5.4 LONG CONTEXT UNDERSTANDING

To further validate the performance of FreqKV on downstream tasks, we SFT LLaMA-2-chat-7b on LongAlpaca (Chen et al., 2024) to extend the context window size from 4K to 8K, and evaluate it on the long context understanding benchmark LongBench (Bai et al., 2024). Scores on the 6 categories of LongBench are reported in Table 3. Detailed results on the 16 tasks can be referred to in Appendix B.

We compare our method with different KV compression strategies, including LM-Infinite (Han et al., 2024), LongHeads (Lu et al., 2024), SnapKV (Li et al., 2024) and PyramidKV (Cai. et al., 2024b). Since the KV cache size of our FreqKV is not fixed and ranges between 2K and 4K during decoding, the cache size used in these baseline methods is set to 4K for comparison. FreqKV achieves SOTA

Table 3: Scores of different KV compression methods on LongBench. The superscript "*" means that we reproduce SanpKV and PyramidKV following their official code for evaluation. The results of LM-Infinite and LongHeads are reported from Lu et al. (2024). The above four methods are evaluated with a cache size of 4K.

| Method | Single-Doc QA | Multi-Doc QA | Summarization | Few-shot Learning | Code | Synthetic | Avg. |
|---|---|---|---|---|---|---|---|
| llama2-chat | 24.90 | 22.60 | 24.70 | 60.00 | 48.10 | 5.90 | 31.0 |
| LM-Infinite | 14.63 | 7.36 | 7.67 | 25.18 | 29.37 | 5.41 | 14.94 |
| LongHeads | 19.45 | 21.42 | 20.59 | 55.80 | 49.04 | **8.03** | 29.06 |
| SnapKV* | 25.39 | 22.52 | 24.62 | **62.92** | 57.70 | 5.34 | 33.08 |
| PyramidKV* | 25.98 | 22.48 | 24.62 | 62.90 | **57.71** | 4.14 | 32.87 |
| FreqKV | **26.70** | **27.10** | **25.54** | 59.97 | 56.02 | 6.66 | **33.67** |

Table 4: FLOPs (TFLOPs) with input sequences of different lengths. We calculate FLOPs for Llama-2-7b with the original attention which leverages full KV states, and with FreqKV with retaining ratios of 0.1, 0.25, 0.5, and 0.75. Experiments are conducted on the ADA6000 GPU of 48GB, where full KV of 10k tokens with float16 will cause an OOM (Out-of-Memory) issue.

| Models | Retaining Ratio | 4K | 6K | 8K | 10K | 12K |
|---|---|---|---|---|---|---|
| Full KV | - | 62.93 | 101.00 | 143.46 | OOM | OOM |
| FreqKV | 0.1 | 62.93 | 92.65 | 125.17 | 155.77 | 187.58 |
| | 0.25 | 62.93 | 93.31 | 124.79 | 157.35 | 187.73 |
| | 0.5 | 62.93 | 94.42 | 125.90 | 157.38 | 188.85 |
| | 0.75 | 62.93 | 94.45 | 125.94 | 157.44 | 188.94 |

(state-of-the-art) on the single-document QA, multi-document QA, and summarization tracks. Our method is also comparable on the other three tracks and obtains the highest average score across all six tracks.

# 6 ABLATION STUDY

## 6.1 RETAINING RATIO

We conduct further studies on the computation cost and performance regarding different retaining ratios. As introduced in Section 4.3, chunk-wise attention is performed for the prefilling tokens. The size of the attention matrix in each chunk is $(N-L-S) \cdot N$ except for the last few tokens. Therefore, the computational cost of self-attention grows approximately linearly with the input length like sliding window attention (Beltagy et al., 2020).

We use torchprofile[1] to count the number of Floating Point Operations (FLOPs) with input sequences of different lengths for LLaMA-2-7b. The statistics given in Table 4 show that FreqKV reduces more FLOPs as the input length grows from 4K to 12K. Despite more compressions will be performed with the retaining ratio ranging from 0.1 to 0.75, the growth in FLOPs is minimal. This is because the compression is performed every $N - L - S$ tokens with the complexity of $O(N\log N)$, which is negligible compared to the quadratic self-attention.

Moreover, we use the validation set of PG-19 to measure the performance and inference overhead of FreqKV with different retaining ratios. The evaluation context length is 8192. Results are shown in Figure 4. While the model performs better in long context language modeling as the retaining ratio increases, the total inference time grows significantly. Although the difference of FLOPs presented in Table 4 is minimal, larger retaining ratios lead to smaller chunk sizes, which determines how

---

[1]https://github.com/zhijian-liu/torchprofile

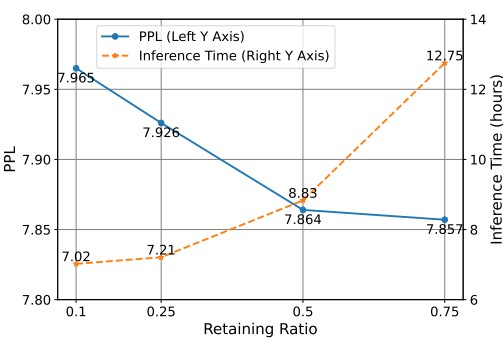 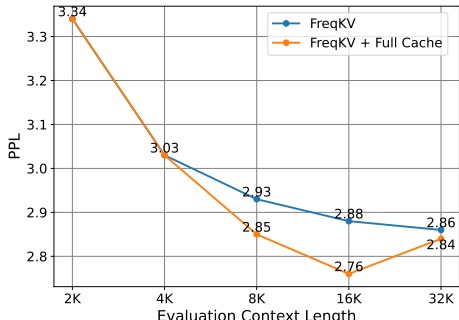

Figure 4: Perplexity evaluation and total inference time on the validation set of PG-19 with different retaining ratios. The evaluation context length is 8192.

Figure 5: Perplexity evaluation on the test set of Proof-pile for FreqKV with full KV cache and compressed cache. The context window of the model is extended to 32K by FreqKV.

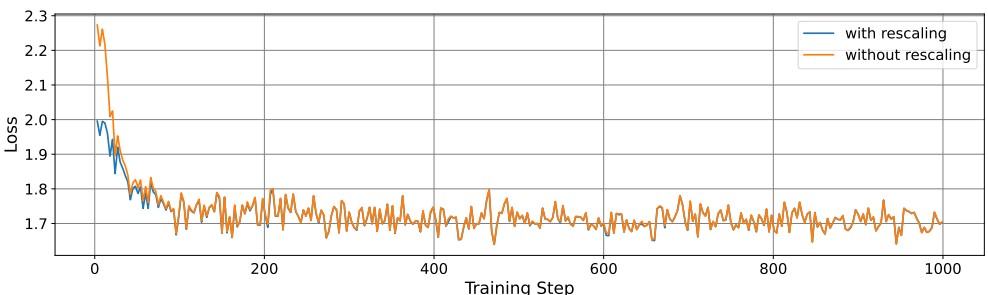

Figure 6: Curves of training loss for FreqKV with and without rescaling.

many attention scores will be masked. As a result, the attention matrix becomes denser and costs an increase in inference overhead. This study justifies our choice of setting a 0.5 retaining ratio as the default for effectiveness and efficiency.

## 6.2 USING FULL KV CACHE DURING INFERENCE

With the model trained with FreqKV, we also evaluate its performance on the test set of Proof-pile when enabling full KV cache during inference. As shown in Figure 5, despite training with the compressed KV on longer contexts, the model achieves better performance when leveraging full KV. It demonstrates that learning the iterative compression of KV states in the frequency domain does not conflict with the original context information.

## 6.3 EFFECT OF RESCALING

In Equation 8, $\sqrt{\frac{L}{N}} = \sqrt{\gamma}$ works as a rescaling factor for the compressed signals to restore the original amplitude when conducting IDCT (the factor is $\sqrt{\frac{L}{N-S}}$ when $S$ sink tokens are uncompressed). To investigate the effect of the rescaling factor, we also use FreqKV to train LLaMA-2-7b without rescaling for comparison. The training curves for FreqKV with and without rescaling are presented in Figure 6.

It can be learned from the figure that, the training loss is significantly higher at the early stages when the compressed signals are not rescaled. This is because IDCT amplifies the compressed states with the normalization factor $\sqrt{\frac{1}{L}}$ progressively in each compression iteration. By rescaling the compressed signals, the training process becomes more stable in its initial phases.

## 7 CONCLUSION

In this paper, we introduce FreqKV to compress KV states iteratively in the frequency domain for LLMs. We exploit the energy concentration of KV states in the frequency domain within the decoder layers. Specifically, we filter out the high-frequency components that are of low magnitude and retain the low-frequency components for compression. The KV cache is compressed in the frequency domain without introducing additional compression modules. Iteratively compressing the KV cache, FreqKV could extend the context window efficiently for LLMs. With minimal training of low-rank adaption, LLMs learn to leverage the compressed KV cache. Through extensive experiments and analysis on long context modeling and understanding, FreqKV demonstrates its efficiency and effectiveness in context extension.

### ETHICS STATEMENTS

Our work pertains to key-value compression and context extension of large language models. In this work, we use only publicly available data and artifacts. There are no ethical issues in our paper, including its motivation and experiments.

### REPRODUCIBILITY STATEMENTS

We have provided detailed implementations of our method throughout the paper. Our method is comprehensively elaborated in Section 4. Detailed settings of our experiments and analyses are given in Section 5 and 6. We will release our code for reproducibility later.

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

## A    HEAD-WISE ANALYSIS OF POWER SPECTRUMS

We also explore the power spectrum distribution of key states and value states in different attention heads of LLaMA-2-7b as shown in Figure 7. Although values of power spectrums vary in different heads, their distribution exhibits similar patterns. They have a consistent tendency to aggregate energy in the low-frequency components along the decoding procedure. It could be promising to study specific differences and associations in different heads or other modules.

## B    DETAILED RESULTS ON LONGBENCH

Table 5: Scores of different KV compression methods on LongBench.

| Method | Single-Document QA | | | Multi-Document QA | | | Summarization | | | Few-shot Learning | | | Code | | Synthetic | |
|---|---|---|---|---|---|---|---|---|---|---|---|---|---|---|---|---|
| | NtrvQA | Qasper | MF-en | HotpotQA | 2WikiMQA | Musique | GovReport | QMSum | MultiNews | TREC | TriviaQA | SAMSum | LCC | RB-P | PCount | PRe |
| llama2-chat | 18.7 | 19.2 | 36.8 | 25.4 | 32.8 | 9.4 | 27.3 | 20.8 | 25.8 | 61.5 | 77.8 | 40.7 | 52.4 | 43.8 | 2.1 | 9.8 |
| LM-Infinite | 0.00 | 18.57 | 25.33 | 27.34 | 31.96 | 7.76 | 11.30 | 2.99 | 8.72 | 32.50 | 29.22 | 13.82 | 34.19 | 24.55 | 5.61 | 5.20 |
| LongHeads | 11.61 | 22.98 | 23.76 | 31.28 | 24.10 | 8.87 | 25.36 | 20.24 | 16.18 | 50.67 | 79.98 | 36.74 | 53.85 | 44.22 | **6.39** | 9.67 |
| SnapKV | **18.78** | 20.68 | **36.70** | 27.83 | 31.51 | 8.21 | **26.92** | 20.68 | 26.25 | **64.00** | 83.26 | **41.49** | **60.70** | 54.69 | 2.92 | 7.75 |
| PyramidKV | 18.44 | 23.09 | 36.41 | 27.43 | 32.11 | 7.90 | 26.83 | 21.02 | 26.02 | **64.00** | 83.26 | 41.45 | 60.58 | 54.83 | 2.03 | 6.25 |
| FreqKV | 17.96 | **27.69** | 34.44 | **35.52** | **34.06** | **11.91** | 26.63 | **22.31** | **27.69** | 55.50 | **83.95** | 40.45 | 56.99 | **55.05** | 2.81 | **10.50** |

Detailed results on the 16 tasks of LongBench are reported in Table 5. FreqKV achieves SOTA on 9 of the 16 long context understanding tasks.

## C    RUSULTS ON NEEDLE-IN-A-HAYSTACK

We report Needle-in-a-Haystack results on LLaMA-2-Chat-7B with an extended context window from 4k to 8k using FreqKV. We also implement the "Local Attention" (Xiong et al., 2022) to extend the context window, which keeps sink tokens and the latest tokens in the cache. It shares the same sink size and retaining size as FreqKV. It is also trained on LongAlpaca using the same setttings. Results of the two extension methods are shown in Figure 8a and Figure 8b. They demonstrate that FreqKV surpasses using a simple local window attention when extending the context window from 4k to 8k.[2]

---

[2]FreqKV does not perform well when the token limit is 4300. This is because the question is around 4k and will be separated under this setting.

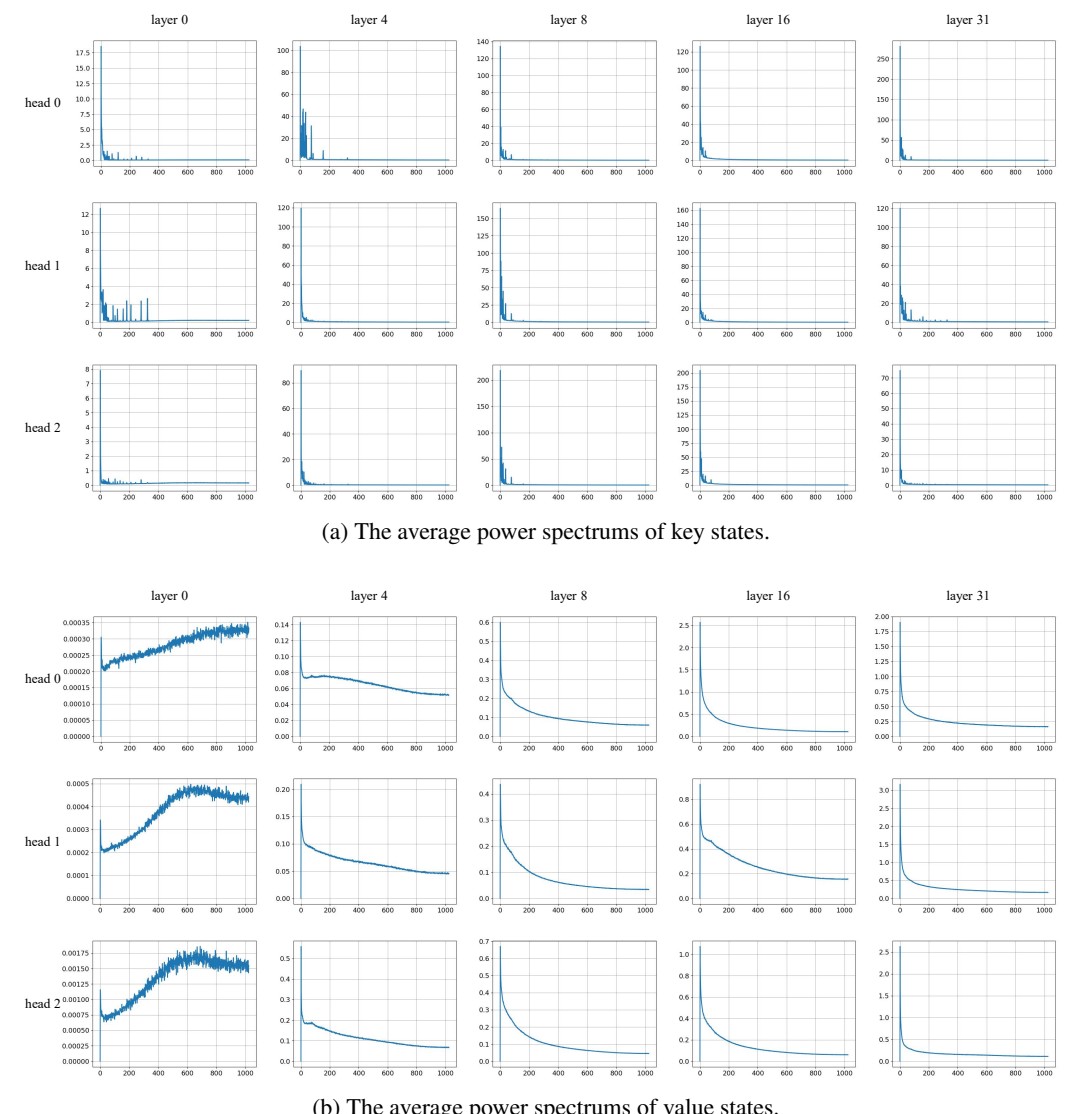

(a) The average power spectrums of key states.

(b) The average power spectrums of value states.

Figure 7: The average power spectrums of key states and value states in different heads of Llama-2-7b.

Moreover, our method achieves an average accuracy of **86.8%**, significantly outperforming the KV compression method SnapKV, which achieves only **49.6%** (Figure 8c) and fails beyond 4k tokens.

## D   COMPRESSION OVERHEAD

To quantify the compression overhead, we have measured FLOPs (TFLOPs) of the "Local Attention". It shares the same sink size and retaining size as FreqKV. The difference in FLOPs between the two methods shows the overhead of compression. The statistics are given in Table 6. The retaining ratio is set to 0.5. The compression times of FreqKV with different context lengths are also reported in the table. It shows that the computation overhead of our compression process grows less than 0.5% even with a length of 16K, which could be negligible.

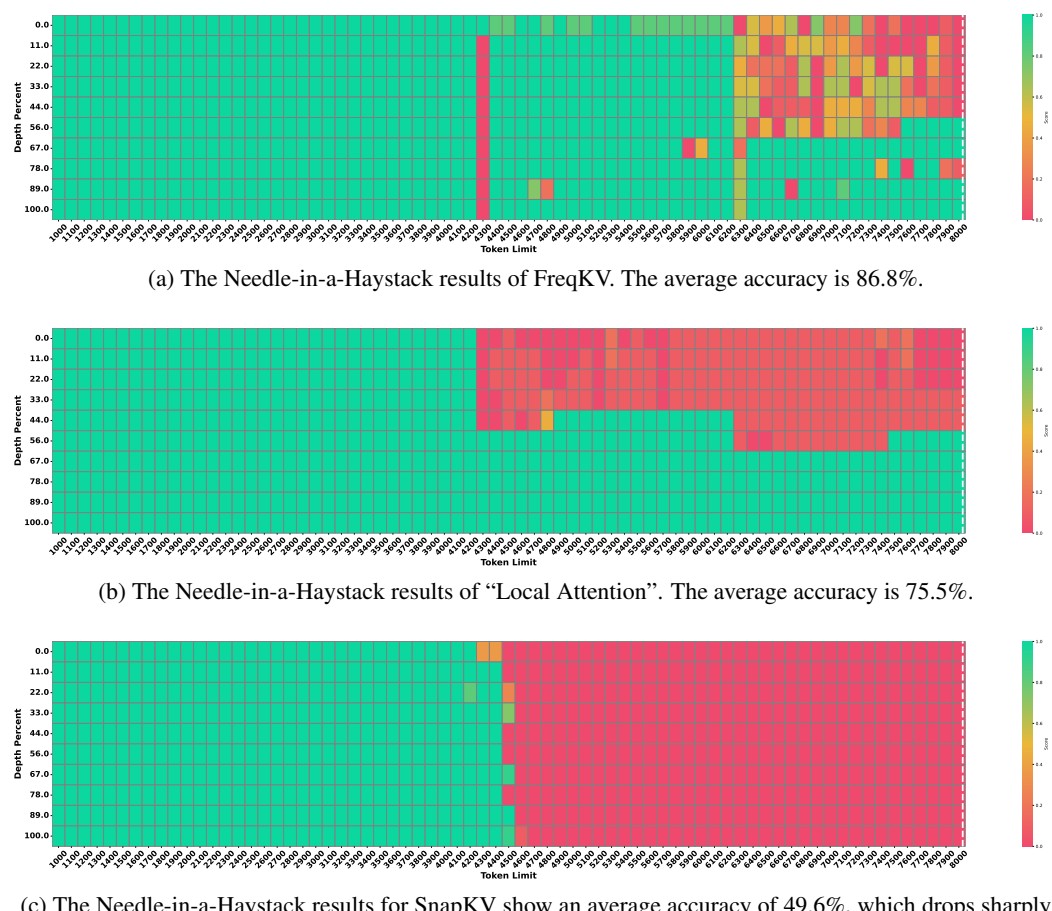

(a) The Needle-in-a-Haystack results of FreqKV. The average accuracy is 86.8%.

(b) The Needle-in-a-Haystack results of "Local Attention". The average accuracy is 75.5%.

(c) The Needle-in-a-Haystack results for SnapKV show an average accuracy of 49.6%, which drops sharply when the context length exceeds LLaMA2's 4k token window.

Figure 8: The Needle-in-a-Haystack resultson LLaMA-2-Chat-7B, with the x-axis representing the document length ("haystack") ranging from 1K to 8K tokens, and the y-axis showing the position of the "needle" (a short sentence) within the document.

Table 6: FLOPs (TFLOPs) with input sequences of different lengths. The difference between FreqKV and "Local Attention" shows the computation overhead of compression.

| Models | 4K | 8K | 12K | 16K |
|---|---|---|---|---|
| Full KV | 62.93 | 143.46 | OOM | OOM |
| Local Attention | 62.93 | 125.86 | 188.79 | 251.72 |
| FreqKV | 62.93 | 125.90 | 188.85 | 251.81 |
| Compression Times | 0 | 3 | 5 | 7 |
| Compression Overhead | 0 (0%) | 0.039 (0.031%) | 0.064 (0.034%) | 0.090 (0.036%) |

# E  PERFORMANCE ON LLAMA3

The original context window length of Llama3-8B-Instruct (AI@Meta, 2024) is 8K. It is equipped with GQA (Grouped-Query Attention), which means it has a lower proportion of parameters for attention modules than Llama2-7B-Base/Chat (Multi-Head Attention, MHA). We use FreqKV to extend the context window from 8K to 16K and evaluate the performance on six Single-Doc QA and Summarization tasks from LongBench. The results of the vanilla model and the SOTA KV compression method SnapKV are also reported in Table 7.

Table 7: Performance of FreqKV on Llama3-8B-Instruct.

| Method | Single-Document QA | | | Summarization | | |
|---|---|---|---|---|---|---|
| | Narrative QA | Qasper | MultiFieldQA-en | GovReport | QMSum | MultiNews |
| Llama3-8B-Instruct | **22.52** | 31.83 | 41.04 | 28.87 | **23.25** | 26.46 |
| SnapKV | 22.27 | 31.93 | 41.03 | 28.77 | 23.14 | 26.62 |
| FreqKV | 20.55 | **32.59** | **45.53** | **30.86** | 22.61 | **28.19** |

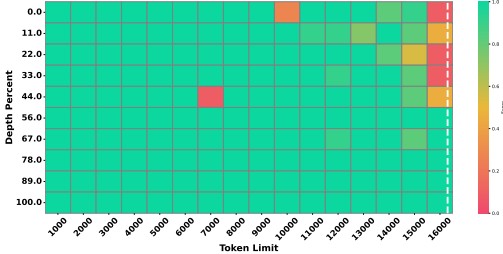 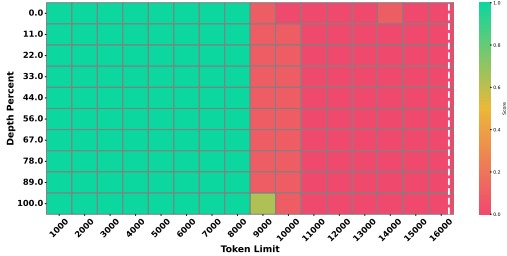

Figure 9: The Needle-in-a-Haystack results of FreqKV on Llama3-8B-Instruct. The average accuracy is 95.2%.

Figure 10: The Needle-in-a-Haystack results of SnapKV on Llama3-8B-Instruct. The average accuracy is 51.5%.

Llama3 and SnapKV are evaluated with a context length of 8K. Their performance drops significantly when setting the context length to 16K. FreqKV is evaluated with a context length of 16K. While SnapKV maintains the performance of the vanilla model, FreqKV enables the model to handle longer contexts and achieves improvements on most tasks.

Moreover, we evaluate FreqKV and SnapKV on Needle-in-a-Haystack with the document length ranging from 1K to 16K as in Figure 9 and Figure 10. FreqKV performs well in extending the context window and achieves an average accuracy of 95.2%, which significantly outperforms SnapKV. These experimental results demonstrate the effectiveness of FreqKV when applied to Llama3-8B-Instruct.

