# OpenReview forum: "FreqKV: Frequency Domain Key-Value Compression for Efficient Context Window Extension"
_ICLR.cc/2025/Conference — Submitted to ICLR 2025_

### Official Review · Reviewer_npHy · 2024-10-23

**Soundness:** 3
**Presentation:** 3
**Contribution:** 4
**Rating:** 6
**Confidence:** 3

**Summary:**

This paper proposes a novel method for compressing the KV cache into a fixed length, by filtering out high-frequency components. This compression can be applied iteratively, so that a small fixed memory budget is enough for processing long contexts. This approach reduces both memory and computational requirements, applicable during both training and inference. The authors demonstrate that this method yields minimal perplexity reduction when compared to full attention (without compression) and compared to other compression methods. They also demonstrate its effectiveness on long context downstream tasks via the LongBench benchmark.

**Strengths:**

**Originality**: The paper presents a novel idea that allows processing longer contexts by making uses of the observation that the information in the KV cache is mostly in the low frequency components. While this observation is not new, using it to compress the KV cache has not been done before, to the best of my knowledge.

**Significance**: As LLMs become stronger and more prevalent, it also becomes more and more important to make them applicable for longer contexts. Therefore, methods like FreqKV that allow processing longer contexts with minimal effect to quality might become invaluable.

**Clarity**: The method is simple, and presented clearly and elegantly.

**Weaknesses:**

**Results:**
* Hard to interpret the strength of the results in table 1 without a comparison to a simple baseline like local attention. While table 1 shows that the higher perplexity is not as bad as in the other compression method (LoCoCo), it would be nice to also show a comparison to local attention. In many cases the difference in perplexity between full attention and local attention might not be very large (e.g., see Xiong et al., 2022, “Simple Local Attentions Remain Competitive for Long-Context Tasks”), so it would be helpful to see if this method of compressing the full context to max size 4k works substantially better than the trivial method of only keeping the latest 4k elements in the KV cache.
* Method’s performance does not strongly exceed competing compression methods such as SnapKV and PyramidKV (table2 shows slightly higher avg for FreqKV but it’s not clear how significant this difference is).

**Interpretation:** while the paper shows that the method works in practice, it does not explain the reasoning behind the observation. Specifically, is there a plausible explanation for why the information in the KV cache is concentrated around low frequency components? And how is the transformer adapting to work with semi-compressed KV cache during fine tuning? While these questions are not crucial for presenting a practical method, discussing them would make the paper stronger.

**Questions:**

* Why is there no overlap between the methods listed in table 3 and table 4? Specifically, why not test the perplexity of SnapKV etc. on PG-19? And why not test LoCoCo on LongBench? Is there any reason why these do not apply?
* As stated in weaknesses - I think it would make the results of table 2 stronger if you include a comparison with a simple baseline like local attention.
* As stated in weaknesses - I think some discussion of the advantages / disadvantages of FreqKV compared to other compression methods (such as SnapKV and PyramidKV) would be helpful. Currently, table 2 shows that these methods seem to be on-par so it’s difficult to understand the advantages of FreqKV without this discussion.

**Small suggestions:**
* Introduction has a typo (“shifted spare attention” --> “shifted sparse attention”)
* In section 4.2 (“KV Compression in the Frequency Domain”), the notation for $\tilde{K}_{0:L-1}^{0:N-1}$ is IMO confusing. Because of the superscript, it took me a while to understand that the shape of $\tilde{K}$ is (L, d) and not (N, d). I think it would be helpful to explain this in the text explicitly.
* Currently both PyramidKV and LoCoCo use the reference “Cai. et al., 2024”.

---

> ### Author Response · Authors · 2024-11-23
> **Response to reviewer npHy**
>
> Dear reviewer, thanks for taking the time to review our paper and for your insightful comments. We will discuss your concerns and refine our paper accordingly.
>
> ## Comparison with other compression methods (Q1, Q3)
>
> **Our work is an efficient context window extension method that implements an iterative KV compression manner. Our intuition is to extend the context window of a pre-trained LLM effectively and efficiently.** The baseline methods in Tab. 1, LongLoRA and LoCoCo, are the most relevant to our work. Our method compresses KV states iteratively to extend the context window and optimizes both fine-tuning and inference efficiency.
>
> From the perspective of KV compression, We compare our method with different strategies like SnapKV and PyramidKV on LongBench. LongBench is the most commonly used for evaluation by these methods. Since our method is capable of compressing KV cache during inference, we compare it with these KV compression methods on LongBench to evaluate language understanding performance and generation quality. However, SnapKV and PyramidKV only compress the KV cache and conduct experiments within the original context window. They are not evaluated on contexts out of the window in their paper. We have tested SnapKV and PyramidKV on PG19 and Proof-pile and got ppl > 100 when extending contexts to 8K on llama2.
>
> As for LoCoCo, they only report an average score on LongBench in their paper and the evaluation script is not given in their official repository. We have implemented it by ourselves and got a much worse performance.
>
> ## Comparison with local attention (Q2)
>
> We have evaluated the performance (PPL) of the Local Attention baseline on Proof-plie with llama2-7b as in the following table. "Local Attention" stands for keeping sink tokens and the latest tokens in the cache. It shares the same sink size and retaining size as FreqKV. The results show that the model benefits from our FreqKV when extending context length.
>
> | evaluation length |  2048  |  4096  |  8192  |
> |-------------------|--------|--------|--------|
> |      Full FT      |  3.14  |  2.85  |  2.66  |
> |  Local Attention  |  3.16  |  2.87  |  2.75  |
> |       FreqKV      |  3.16  |  2.88  |  2.70  |
>
> ## Further Interpretation of our work
>
> As introduced in Sec. 1, we are inspired by the well-established technique to compress images in the frequency domain in CV. It has been observed that low-frequency channels are more important for CNNs. It occurs to us whether the hidden states of texts can be compressed in the frequency domain. We transform key
> states and value states in LLaMA-2-7b from the time domain to the frequency
> domain for power spectrum analysis. As shown in Figure 1, the energy distribution increasingly concentrates on low-frequency components as the computation process progresses inside the model. This suggests that high-frequency components, which contribute less to the overall information, can be discarded without a significant impact on performance, thereby enhancing computational efficiency.
>
> Previous work in NLP has studied to train Transformer encoder architectures in the frequency domain like FNet and Fourier Transformer. Our work aims to extend the context window efficiently by compressing key-value states in the frequency domain for decoder-only LLMs. For contexts out of the original window size, the model only uses the compressed KV. Additional training adapts the model to leverage the new pattern of KV when extending to longer contexts.
>
> ## Regarding all the typos you pointed out
>
> Thank you for your suggestions. We will revise them in the paper.
>
> We hope that the provided analysis and discussion have addressed your concerns.

---

> > ### Comment · Reviewer_npHy · 2024-11-24
> >
> > Thank you for your clarifications! Regarding the table you provided - I'm a bit confused about the perplexity for FreqKV on "evaluation context length"=8192 with FreqKV. In your manuscript (table 1), the value for FreqKV/8192 is 2.80, but in the table you provided above it is 2.70. Which is the correct number?

---

> > > ### Author Response · Authors · 2024-11-24
> > > **Comparison with Local Attention**
> > >
> > > Thank you for the reminder. The correct number is 2.80 as in the paper. Since the difference in ppl is minimal, we conduct experiments on the Needle-in-a-Haystack task. FreqKV achieves an average accuracy of 86.8%, outperforming Local Attention, which achieves only 75.5%. Local Attention suffers from performance degradation when extending the context window from 4k to 8k. Detailed results are provided in the appendix of the revised version.

---

> > > > ### Comment · Reviewer_npHy · 2024-11-24
> > > >
> > > > Thank you for the detailed response and clarifications. I think the method is really interesting and shows promise, but due to the results not being very strong, I am keeping my rating at marginally above acceptance.

---

> > > > > ### Author Response · Authors · 2024-11-30
> > > > > **Performance on Llama3**
> > > > >
> > > > > Dear reviewer,
> > > > >
> > > > > We have conducted experiments on Llama3-8B-Instruct to validate its effectiveness on other models.
> > > > >
> > > > > The original context window length of Llama3-8B-Instruct is 8K. It is equipped with GQA (Grouped-Query Attention), which means it has a lower proportion of parameters for attention modules than Llama2-7B-Base/Chat (MHA). We use FreqKV to extend the context window from 8K to 16K and evaluate the performance on six Single-Doc QA and Summarization tasks from LongBench. The results of the vanilla model and the SOTA KV compression method SnapKV are also reported in the following table.
> > > > >
> > > > > |  models  |Narrative QA|Qasper|MultiFieldQA-en|GovReport|QMSum|MultiNews|
> > > > > |--|--|--|--|--|--|--|
> > > > > |Llama3-8B-Instruct|**22.52**|31.83|41.04|28.87|**23.25**|26.46|
> > > > > |+ SnapKV|22.27|31.93|41.03|28.77|23.14|26.62|
> > > > > |+ FreqKV|20.55|**32.59**|**45.53**|**30.86**|22.61|**28.19**|
> > > > >
> > > > > Llama3 and SnapKV are evaluated with a context length of 8K. Their performance drops significantly when setting the context length to 16K. FreqKV is evaluated with a context length of 16K. While SnapKV maintains the performance of the vanilla model, FreqKV enables the model to handle longer contexts and achieves improvements on most tasks. It demonstrates the effectiveness of FreqKV when applied to Llama3-8B-Instruct.
> > > > >
> > > > > Moreover, we evaluate FreqKV and SnapKV on **Needle-in-a-Haystack** with the document length ranging from 1K to 16K. FreqKV performs well in extending the context window and achieves an average accuracy of 95.2%, significantly outperforming SnapKV which achieves only 51.5%.
> > > > >
> > > > > **Detailed results are provided in the Appendix. E of the revised version**. These experimental results demonstrate the effectiveness of FreqKV when applied to Llama3-8B-Instruct.
> > > > >
> > > > > We hope that the provided results on Llama3 have addressed your concerns.

---

> > > > > ### Author Response · Authors · 2024-12-02
> > > > >
> > > > > Dear Reviewer,
> > > > >
> > > > > Thank you for the time and attention you've dedicated to reviewing our paper. As the discussion stage is coming to an end, please let us know whether we have addressed your concerns or if any further discussions are needed.
> > > > >
> > > > > We look forward to your feedback and thank you once again for your valuable contribution to our work.
> > > > >
> > > > > Best, Authors

---

> > > > > > ### Comment · Reviewer_npHy · 2024-12-03
> > > > > >
> > > > > > Thank you for providing the additional needle-in-a-heystack experiment. However, I still think the results presented in the main paper (specifically table 1 and table 3) are not very strong. In table 1 my main concern was that a simple method like local attention might get similar results to FreqKV and this was confirmed with the results of the experiment provided by the authors during the review (although these results were not added to table 1 of the paper which would make this observation harder to see for future readers).
> > > > > >
> > > > > > I still think it is a good paper, but I do not believe it’s strongly passes the accept threshold, which is why I am not changing my current score.

---

> > > > > > > ### Author Response · Authors · 2024-12-03
> > > > > > > **Further discussion about FreqKV and Local Attention**
> > > > > > >
> > > > > > > Dear reviewer,
> > > > > > >
> > > > > > > Thank you for your response. Although the difference in ppl between FreqKV and Local Attention is minimal, **FreqKV performs much better in the Needle-in-a-Haystack task**. As shown in Figure 8 (Appendix C), FreqKV achieves an average accuracy of 86.8%, outperforming Local Attention, which achieves only 75.5%. Despite training on LongAlpaca using the same settings, Local Attention evicts tokens directly and suffers from performance degradation when extending the context window. However, FreqKV mixes tokens when conducting DCT and IDCT as shown in the formula of Equations 1 and 2. It will not lose the information associated with evicted tokens like Local Attention.

---

### Official Review · Reviewer_V44X · 2024-10-27

**Soundness:** 3
**Presentation:** 4
**Contribution:** 4
**Rating:** 8
**Confidence:** 5

**Summary:**

This work introduces a novel context extension training method that compresses key-value states in the frequency domain rather than in the time domain. By analyzing the KV Cache in each layer and head of the Llama-2 model in the frequency domain, they found that the energy is concentrated in the low-frequency components. Based on this observation, they retain only specific low-frequency signals from the expanded KV Cache during model fine-tuning and explore various compression strategies. The proposed method aims to balance runtime speedup with an improved trade-off between model performance and loss.

**Strengths:**

1. The work brings significant novelty by providing a detailed analysis of why frequency-domain compression is suitable, based on layer-by-layer observations in the Llama model. This analysis leads to the proposal that caching only low-frequency components of the KV states is sufficient. As the author points out, previous approaches that evict KV Cache lead to a permanent loss of information, especially when extending sequence length, whereas this method mitigates that issue.

2. The theoretical analysis is robust, particularly in sections 3.1, 3.2, and 4.1, providing a solid foundation for the proposed approach.

3. The method delivers strong performance in both accuracy and latency. The ablation studies, particularly on sink tokens and the choice of $L$, further reinforce the approach's effectiveness.

**Weaknesses:**

1. The study is somewhat limited in scope, as it focuses on a single model, making it difficult to generalize the approach as a universal method for all decoder-only generative LLMs.

2. The benchmarks for long-text sequences are relatively few, which may limit the comprehensive evaluation of the method's effectiveness in handling extended sequences.

**Questions:**

1. You've only tested on Llama-2-7B-(chat). I'm curious about how the proposed "cache low-frequency is enough" approach would perform on other models such as Mistral or Llama-3/3.1 (GQA). If it can demonstrate similar advantages on these models, it would be incredibly exciting and broaden the impact of the work.

2. If possible, I’d like to see how this method performs on benchmarks like **Ruler** (https://github.com/zhang677/RULER) and **Needle In A Haystack** (https://github.com/gkamradt/LLMTest_NeedleInAHaystack/). These could provide a more diverse and challenging evaluation of the approach.

3. If these points can be addressed, I will reconsider the score.

---

> ### Author Response · Authors · 2024-11-23
> **Response to reviewer V44X**
>
> Dear reviewer, thanks for taking the time to review our paper and for your insightful comments. We will discuss your concerns and refine our paper accordingly.
>
> ## Choice of the model (Q1)
>
> FreqKV is an efficient context window extension method that implements an iterative KV compression manner. Our intuition is to extend the context window of a pre-trained LLM effectively and efficiently. Most related works (LongLoRA, LoCoCo, and Activation Beacon) use llama2 as the base model to extend its context window. Following these studies, we use llama2 to evaluate the performance of context extension and long-context understanding. Models after llama2 have already been trained to support longer contexts (32k for Mistral and 128k for LLaMA3). Extending their context length requires more resources than we currently have, so we are not able to conduct experiments on these models.
>
> ## Experiments on other benchmarks (Q2)
>
> We find Ruler is a very recent benchmark and is not used for evaluation by the baseline methods in our paper. Due to time and resource constraints, we choose the Needle-in-a-Haystack task, aligning with most prior methods. FreqKV achieves an accuracy of 86.8%, significantly outperforming SnapKV's 49.6%. Unlike FreqKV, which extends the context window while compressing the KV cache, SnapKV focuses solely on KV compression, leading to an accuracy drop to 0 when the test context exceeds the original window size. Detailed results are provided in the appendix of the revised version.
>
> We hope that the provided analysis and discussion have addressed your concerns.

---

> > ### Comment · Reviewer_V44X · 2024-11-25
> > **Adjust my score to 6**
> >
> > Thank you for your comments. The results in NIAH are satisfactory for me. However, I noticed that the Llama-3-8B model appears to have a maximum context length of 8k not 32k, as indicated https://huggingface.co/meta-llama/Meta-Llama-3-8B-Instruct/blob/main/config.json#L13. Why not base on llama3 to do long-conext extension? Generally speaking, Llama-3 has made significant improvements compared to Llama-2 (Llama-2 is easier to compress, quantize, or sparsify while maintaining relatively high accuracy).
> >
> > I also share some concerns about whether this method can achieve good results, even though the innovation behind the approach is commendable. Based on this, I am inclined to adjust my score to 6.

---

> > > ### Author Response · Authors · 2024-11-30
> > > **Performance on Llama3**
> > >
> > > Dear reviewer,
> > >
> > > As you are interested in the performance of FreqKV on other models, we have conducted experiments on Llama3-8B-Instruct to validate its effectiveness. "128k for LLaMA3" in our previous response stands for llama3.1/3.2. Sorry for the confusion.
> > >
> > > The original context window length of Llama3-8B-Instruct is 8K. It is equipped with GQA (Grouped-Query Attention), which means it has a lower proportion of parameters for attention modules than Llama2-7B-Base/Chat (MHA). We use FreqKV to extend the context window from 8K to 16K and evaluate the performance on six Single-Doc QA and Summarization tasks from LongBench. The results of the vanilla model and the SOTA KV compression method SnapKV are also reported in the following table.
> > >
> > > |  models  |Narrative QA|Qasper|MultiFieldQA-en|GovReport|QMSum|MultiNews|
> > > |--|--|--|--|--|--|--|
> > > |Llama3-8B-Instruct|**22.52**|31.83|41.04|28.87|**23.25**|26.46|
> > > |+ SnapKV|22.27|31.93|41.03|28.77|23.14|26.62|
> > > |+ FreqKV|20.55|**32.59**|**45.53**|**30.86**|22.61|**28.19**|
> > >
> > > Llama3 and SnapKV are evaluated with a context length of 8K. Their performance drops significantly when setting the context length to 16K. FreqKV is evaluated with a context length of 16K. While SnapKV maintains the performance of the vanilla model, FreqKV enables the model to handle longer contexts and achieves improvements on most tasks. It demonstrates the effectiveness of FreqKV when applied to Llama3-8B-Instruct.
> > >
> > > Moreover, we evaluate FreqKV and SnapKV on **Needle-in-a-Haystack** with the document length ranging from 1K to 16K. FreqKV performs well in extending the context window and achieves an average accuracy of 95.2%, significantly outperforming SnapKV which achieves only 51.5%.
> > >
> > > **Detailed results are provided in the Appendix. E of the revised version**. These experimental results demonstrate the effectiveness of FreqKV when applied to Llama3-8B-Instruct.
> > >
> > > We hope that the provided results on Llama3 have addressed your concerns.

---

> > > ### Author Response · Authors · 2024-12-02
> > >
> > > Dear Reviewer,
> > >
> > > Thank you for the time and attention you've dedicated to reviewing our paper. As the discussion stage is coming to an end, please let us know whether we have addressed your concerns or if any further discussions are needed.
> > >
> > > We look forward to your feedback and thank you once again for your valuable contribution to our work.
> > >
> > > Best, Authors

---

> > > > ### Comment · Reviewer_V44X · 2024-12-02
> > > >
> > > > Thank you for the llama-3-8B results. It actually answers my questions. I will raise my score.

---

> > > > > ### Author Response · Authors · 2024-12-03
> > > > >
> > > > > We truly appreciate your recognition and support! Thank you once again for the time and diligence you've dedicated to reviewing our paper.

---

### Official Review · Reviewer_RuV5 · 2024-11-01

**Soundness:** 2
**Presentation:** 3
**Contribution:** 2
**Rating:** 5
**Confidence:** 4

**Summary:**

This paper presents FreqKV which uses the discrete cosine transform to compress the KV cache in the frequency domain by removing the long tail of high frequency components, then uses iDCT on the fly to decompress the KV cache and use it normally within the model. The first few tokens are not compressed, similar to prior work on Attention Sinks. Compression is performed iteratively, such that whenever a new set of KV tokens are produced, it is compressed together with the previously-compressed tokens. This results in a linear increase in decoding time instead of quadratic.

**Strengths:**

KV-cache compression is an important open problem and frequency-domain transform is an interesting method that could be effective. Freq domain is also an interesting space for compression because computation can be done in the frequency domain as was previously demonstrated with CNNs. This work does not take advantage of this feature though.

**Weaknesses:**

1- What is the overhead of DCT and iDCT on the fly in every iteration? This does not seem to be factored in the performance measurements that you performed in the paper. How do you perform these transforms?
2- The presented method is a composition of attention sinks and freq-domain compression, but the baseline attention sinks result is not shown.
3- More evaluation on long-context results would strengthen the case. Since this is a KV-cache compression method, I find ppl results somewhat irrelevant. Longbench results are good, but I suggest adding GSM8k, needle-in-haystack, and other purpose-built benchmarks.
4- I didn't fully get why you needed to extend llama2 context to 32k instead of using llama3, which is already longer context (128k)?

**Questions:**

see weaknesses - most of my questions are there.

---

> ### Author Response · Authors · 2024-11-23
> **Response to reviewer RuV5**
>
> Dear reviewer, thanks for taking the time to review our paper and for your insightful comments. We will discuss your concerns and refine our paper accordingly.
>
> ## Compression overhead (Q1)
>
> As introduced in Sec. 4.2, KV is compressed in the frequency domain. The whole compression process consists of DCT, filtration of high-frequency components, and IDCT. KV is transferred to the frequency domain by DCT, and then compressed. IDCT is used to convert the frequency components back to the time domain. (Line 202) It does not increase the size of the compressed KV.
>
> Our compression occurs on-the-fly during inference. As illustrated in Sec. 4.3, when the cache is not filled, subsequent tokens could get in and fill the cache. When the cache is filled, it will be compressed so that more tokens could get in until the cache is filled again.
>
> We maintain S = 4 sink tokens uncompressed. The retaining ratio γ in compression is set to 0.5. Therefore, the retaining size during each compression is L = γ · (N − S) = 2046. As long as the cache size reaches its capacity of N = 4096, the 4092 states since the 5-th state in the cache will be compressed into 2046 states. The compression is performed every N − L − S tokens with the complexity of O(NlogN).
>
> To quantify compression overhead, we have measured FLOPs (TFLOPs) with different context lengths on llama2-7b. "Sink+Recent" stands for keeping sink tokens and the latest tokens in the cache. It shares the same sink size and retaining size as FreqKV. The difference in FLOPs between the two methods shows the overhead of compression. The statistics are given in the following table. It shows that the computation overhead of our compression process grows less than 0.5% even with a length of 16K and could be negligible.
>
> |   context length   |   4K   |   8K   |   12K   |   16K   |
> |--------------------|--------|--------|---------|---------|
> |      Full KV       | 62.93  | 143.46 |   OOM   |   OOM   |
> |     Sink+Recent    | 62.93  | 125.86 | 188.79  |  251.72 |
> |       FreqKV       | 62.93  | 125.90 | 188.85  |  251.81 |
> |Compression Overhead| 0 (0%) |0.039 (0.031%)|0.064 (0.034%)|0.090 (0.036%)|
> | Compression Times  |    0   |   3    |    5    |   7   |
>
> ## Study of attention sinks (Q2)
>
> We have evaluated the performance (PPL) of the attention sink baseline on Proof-plie as in the following table. "Sink+Recent" stands for keeping sink tokens and the latest tokens in the cache. It shares the same sink size and retaining size as FreqKV. The results show that the model benefits from our FreqKV when extending context length.
>
> | evaluation length |  2048  |  4096  |  8192  |
> |-------------------|--------|--------|--------|
> |      Full FT      |  3.14  |  2.85  |  2.66  |
> |    Sink+Recent    |  3.16  |  2.87  |  2.75  |
> |       FreqKV      |  3.16  |  2.88  |  2.70  |
>
> ## More long-context benchmarks (Q3)
>
> We have evaluated FreqKV on the Needle-in-a-Haystack task and compared it with SnapKV. FreqKV achieves an accuracy of 86.8%, significantly outperforming SnapKV's 49.6%. Unlike FreqKV, SnapKV is solely a KV compression method and cannot extend the LLMs' context window, causing its accuracy to drop to 0 when the context exceeds the original window. Detailed results are provided in the appendix of the revised version.
>
> Since GSM8k is a math reasoning benchmark with question lengths under 1k tokens, it is not considered a long-context benchmark, and we do not evaluate our method on it.
>
> ## Why llama2 but not llama3 (Q4)
>
> FreqKV is an efficient context window extension method that implements an iterative KV compression manner. Our intuition is to extend the context window of a pre-trained LLM effectively and efficiently. Most related works (LongLoRA, LoCoCo, and Activation Beacon) use llama2 as the base model to extend its context window. Following these studies, we use llama2 to evaluate the performance of context extension and long-context understanding. Models after llama2 have already been trained to support longer contexts (32k for Mistral and 128k for LLaMA3). Extending their context length requires more powerful GPUs and is more difficult to evaluate.
>
> We hope that the provided analysis and discussion have addressed your concerns.

---

> > ### Comment · Reviewer_RuV5 · 2024-11-23
> > **Thank you for the response**
> >
> > The authors helped to clarify some of my questions. The method is intresting, although it is confusing sometimes to compare to KV cache compression methods, and in other times to KV-cache extension methods. To me, FreqKV should be evaluated as a KV-cache compression method - this is why I asked about Llama2 vs Llama3. For context extension, thanks to the reviewers for showing the sinks+window baseline at different context lengths, indeed showing a perplexity improvement (though very small) at longer contexts. I think a. more appropriate comparison would be to some long context task (for e.g. including attention sinks in the response to Q3 above), but the table shown in the rebuttal is already showing a positive signal.
> >
> > Based on the above, and the fact that the results are not very strong, I will maintain my borderline review of the work.

---

> > > ### Author Response · Authors · 2024-11-24
> > > **Comparison with attention sinks**
> > >
> > > Thank Reviewer npHy for the reminder that the ppl for FreqKV/8192 is 2.80. Since the difference in ppl is minimal, we conduct experiments on the Needle-in-a-Haystack task. FreqKV achieves an average accuracy of 86.8%, outperforming Local Attention, which achieves only 75.5%. Local Attention suffers from performance degradation when extending the context window from 4k to 8k. Detailed results are provided in the appendix of the revised version.

---

> > > ### Author Response · Authors · 2024-12-02
> > >
> > > Dear Reviewer,
> > >
> > > Thank you for the time and attention you've dedicated to reviewing our paper. As the discussion stage is coming to an end, please let us know whether we have addressed your concerns or if any further discussions are needed.
> > >
> > > We look forward to your feedback and thank you once again for your valuable contribution to our work.
> > >
> > > Best, Authors

---

> ### Author Response · Authors · 2024-11-30
> **Performance on Llama3**
>
> Dear reviewer,
>
> As you are interested in the performance of FreqKV on llama3, we have conducted experiments on Llama3-8B-Instruct to validate its effectiveness. Llama3.1/3.2 have already been trained to support longer contexts (128K). Extending their context length requires more resources than we currently have, so we are not able to conduct experiments on these models.
>
> The original context window length of Llama3-8B-Instruct is 8K. It is equipped with GQA (Grouped-Query Attention), which means it has a lower proportion of parameters for attention modules than Llama2-7B-Base/Chat (MHA). We use FreqKV to extend the context window from 8K to 16K and evaluate the performance on six Single-Doc QA and Summarization tasks from LongBench. The results of the vanilla model and the SOTA KV compression method SnapKV are also reported in the following table.
>
> |  models  |Narrative QA|Qasper|MultiFieldQA-en|GovReport|QMSum|MultiNews|
> |--|--|--|--|--|--|--|
> |Llama3-8B-Instruct|**22.52**|31.83|41.04|28.87|**23.25**|26.46|
> |+ SnapKV|22.27|31.93|41.03|28.77|23.14|26.62|
> |+ FreqKV|20.55|**32.59**|**45.53**|**30.86**|22.61|**28.19**|
>
> Llama3 and SnapKV are evaluated with a context length of 8K. Their performance drops significantly when setting the context length to 16K. FreqKV is evaluated with a context length of 16K. While SnapKV maintains the performance of the vanilla model, FreqKV enables the model to handle longer contexts and achieves improvements on most tasks. It demonstrates the effectiveness of FreqKV when applied to Llama3-8B-Instruct.
>
> Moreover, we evaluate FreqKV and SnapKV on **Needle-in-a-Haystack** with the document length ranging from 1K to 16K. FreqKV performs well in extending the context window and achieves an average accuracy of 95.2%, significantly outperforming SnapKV which achieves only 51.5%.
>
> **Detailed results are provided in the Appendix. E of the revised version**. These experimental results demonstrate the effectiveness of FreqKV when applied to Llama3-8B-Instruct.
>
> We hope that the provided results on Llama3 have addressed your concerns.

---

### Official Review · Reviewer_jzk2 · 2024-11-03

**Soundness:** 3
**Presentation:** 3
**Contribution:** 2
**Rating:** 5
**Confidence:** 4

**Summary:**

The authors introduce "FreqKV: Frequency Domain Key-Value Compression for Efficient Context Window Extension," a novel approach aimed at reducing the computational and memory demands associated with the KV-cache in large language models (LLMs) during both training and inference. This method uses a frequency-domain compression technique that retains only the low-frequency components of the KV cache, with the goal of optimizing memory usage and computational complexity for extended context windows.

**Strengths:**

1) Problem importance: FreqKV addresses the critical problem of context window extension in LLMs, specifically targeting issues of performance degradation, high computational complexity, and excessive memory consumption as context length increases which is an important challenge for enabling LLMs to handle long-form content efficiently in real-world applications.

2) Novelty of the approach: FreqKV introduces a novel approach by using frequency-based compression for the first time in context extension, compressing the information of all tokens instead of fully discarding some, as many other methods do.

3) Training and inference efficiency: FreqKV achieves efficient training and inference without introducing architectural changes or extra parameters. In both training and inference, memory usage is limited by a fixed cache size, and computational complexity grows linearly compared to the quadratic growth in original LLMs. This significantly improves latency and reduces memory consumption, especially for handling longer context windows.

4) Results: FreqKV outperforms the different KV compression techniques on LongBech in three tasks of Single Doc QA, Multi Doc QA, and Summarization.

**Weaknesses:**

1) Performance degradation in training: According to Table 1 of the paper, FreqKV shows higher perplexity compared to LongLoRA, particularly for context lengths of 4096 tokens or more on both test sets. This indicates that FreqKV may underperform slightly in language modeling accuracy at extended context lengths.

2) Results on higher context length: Table 1 reports results for context lengths up to 32K tokens, with no further results on longer contexts such as 128K. This leaves the method’s performance on very large context lengths untested and unclear.

3) Resource usage: While FreqKV does not introduce additional parameters or architectural changes, it still requires extra computational resources for the compression process. The computational complexity and latency overhead are negligible, as compression only occurs when the cache is filled.

4) Results of the other models: The results were reported only on the LLaMA-2-7b model (both base and chat versions), leaving FreqKV's performance on other LLMs unclear.

5) Generalizability:  In the paper, the optimal retaining ratio for FreqKV is determined through an ablation study. Applying this approach to other large language models could be time-consuming, as the best retaining ratio may vary from one model to another.

6) Unifrom across layers: A limitation of FreqKV is its use of a uniform retaining ratio and cache size across all layers. Previous works (such as [1, 2, 3]) have shown that middle layers are particularly important for retrieval and reasoning tasks, suggesting that the importance of each layer's KV cache can vary depending on the task and model. Hence, some layers may contain more important information and would benefit from fewer rounds of compression.

[1] Wenhao Wu, Yizhong Wang, Guangxuan Xiao, Hao Peng, and Yao Fu. Retrieval head mechanistically
explains long-context factuality. arXiv preprint arXiv:2404.15574, 2024.

[2] Yao Fu. How do language models put attention weights over long context. Yao FuâA˘Zs Notion ´ , 2024.

[3] Zhenyu Zhang, Ying Sheng, Tianyi Zhou, Tianlong Chen, Lianmin Zheng, Ruisi Cai, Zhao Song,
Yuandong Tian, Christopher Ré, Clark W. Barrett, Zhangyang Wang, and Beidi Chen. H2O:
heavy-hitter oracle for efficient generative inference of large language models. In Alice Oh,
Tristan Naumann, Amir Globerson, Kate Saenko, Moritz Hardt, and Sergey Levine (eds.),
Advances in Neural Information Processing Systems 36: Annual Conference on Neural Information Processing Systems 2023, NeurIPS 2023, New Orleans, LA, USA, December 10 -
16, 2023, 2023. URL http://papers.nips.cc/paper_files/paper/2023/hash/6ceefa7b15572587b78ecfcebb2827f8-Abstract-Conference.html.

**Questions:**

1) How would the results of FreqKV change if applied to context lengths greater than 32K tokens? Would any additional modifications be necessary in FreqKV to support such extended context lengths?

2) What are the results of FreqKV on other popular large language models, such as GPT models or Gemini? How would the results of FreqKV change when applied to retrieval or reasoning tasks(such as Need-in-a-Haystack)?

4) Is there an algorithmic or automatic method for determining the optimal retaining ratio for FreqKV rather than relying on manual selection through ablation studies?

5) How would the model's performance change if non-uniform cache sizes and retention ratios were used across different layers? Would an experiment comparing uniform compression to a non-uniform approach (where compression rates vary by layer) show the benefits of a more flexible compression strategy?

---

> ### Author Response · Authors · 2024-11-23
> **Response to reviewer jzk2 (part1)**
>
> Dear reviewer, thanks for taking the time to review our paper and for your insightful comments. We will discuss your concerns and refine our paper accordingly.
>
> ## Comparison with LongLoRA (W1)
>
> Our intuition is to extend the context window of a pre-trained LLM effectively and efficiently. LongLoRA uses sparse attention to reduce the training FLOPs when extending to a longer context. **However, it still requires the original attention on the full sequence during inference.** The results of LongLoRA reported in Tab. 1 are using full KV of the sequence during evaluation. If they still use the sparse attention during inference, their performance drops significantly. On the contrary, our method compresses KV states iteratively to extend the context window and optimizes both fine-tuning and inference efficiency. We use the compressed KV during evaluation.
>
> ## Compression overhead (W3)
>
> Our compression occurs on-the-fly during inference. As illustrated in Sec. 4.3, when the cache is not filled, subsequent tokens could get in and fill the cache. When the cache is filled, it will be compressed so that more tokens could get in until the cache is filled again.
>
> We maintain S = 4 sink tokens uncompressed. The retaining ratio γ in compression is set to 0.5. Therefore, the retaining size during each compression is L = γ · (N − S) = 2046. As long as the cache size reaches its capacity of N = 4096, the 4092 states since the 5-th state in the cache will be compressed into 2046 states. The compression is performed every N − L − S tokens with the complexity of O(NlogN).
>
> To quantify compression overhead, we have measured FLOPs (TFLOPs) with different context lengths. "Sink+Recent" stands for keeping sink tokens and the latest tokens in the cache. It shares the same sink size and retaining size as FreqKV. The difference in FLOPs between the two methods shows the overhead of compression. The statistics are given in the following table. It shows that the computation overhead of our compression process grows less than 0.5% even with a length of 16K and could be negligible.
>
> |   context length   |   4K   |   8K   |   12K   |   16K   |
> |--------------------|--------|--------|---------|---------|
> |      Full KV       | 62.93  | 143.46 |   OOM   |   OOM   |
> |     Sink+Recent    | 62.93  | 125.86 | 188.79  |  251.72 |
> |       FreqKV       | 62.93  | 125.90 | 188.85  |  251.81 |
> |Compression Overhead| 0 (0%) |0.039 (0.031%)|0.064 (0.034%)|0.090 (0.036%)|
> | Compression Times  |    0   |   3    |    5    |   7   |

---

> ### Author Response · Authors · 2024-11-23
> **Response to reviewer jzk2 (part2)**
>
> ## Extension to longer contexts (Q1)
>
> To extend to longer contexts (like > 32K) without additional modifications, the only necessity might be GPUs with larger memory. As shown in Tab. 2, the memory usage for each GPU is 44.90GB. We are not able to extend our method to longer contexts directly since we only have 48GB-ADA6000 and 24GB-RTX4090.
>
> However, one of the possible solutions is model quantization, which employs low-bit integer (INT) and low-bit floating-point (FP) formats [1,2]. Another direction is model pruning, which removes part of the model parameters [3,4]. These methods are orthogonal to context extension or compression and can be combined together for specific requirements.
>
> [1] [Integer or Floating Point? New Outlooks for Low-Bit Quantization on Large Language Models](https://arxiv.org/abs/2305.12356)
>
> [2] [AWQ: Activation-aware Weight Quantization for LLM Compression and Acceleration](https://arxiv.org/abs/2306.00978)
>
> [3] [LLM-Pruner: On the Structural Pruning of Large Language Models](https://arxiv.org/abs/2305.11627)
>
> [4] [SliceGPT: Compress Large Language Models by Deleting Rows and Columns](https://arxiv.org/abs/2401.15024)
>
> ## Choice of the model (Q2)
>
> FreqKV is an efficient context window extension method that implements an iterative KV compression manner. Our intuition is to extend the context window of a pre-trained LLM effectively and efficiently. Most related works (LongLoRA, LoCoCo, and Activation Beacon) use llama2 as the base model to extend its context window. Following these studies, we use llama2 to evaluate the performance of context extension and long-context understanding. Models after llama2 have already been trained to support longer contexts (32k for Mistral and 128k for LLaMA3). Extending their context length requires more resources than we currently have, so we are not able to conduct experiments on these models.
>
> ## Experiments on Needle-in-a-Haystack (Q2)
>
> We have evaluated FreqKV on the Needle-in-a-Haystack task and compared it with SnapKV. FreqKV achieves an accuracy of 86.8%, significantly outperforming SnapKV's 49.6%. Unlike FreqKV, SnapKV is solely a KV compression method and cannot extend the LLMs' context window, causing its accuracy to drop to 0 when the context exceeds the original window. Detailed results are provided in the appendix of the revised version.
>
> ## Adaptive retaining ratios across layers (Q4)
>
> We agree with you that different layers utilize KV in different patterns. As shown in Fig. 1, the energy of key states and value states is increasingly concentrated in the low-frequency components as the layer gets deeper. It might further improve FreqKV if we retain more cache in shallow layers and less cache in deep layers. Moreover, we can even use different retaining ratios in different heads as mentioned in Appendix A.
>
> For the clarity of our method, we do not introduce an adaptive method for determining retaining ratios across layers. Most of the long-context compression methods can be equipped with an adaptive method to determine hyper-parameters. It could be promising to study specific differences and associations across different layers, heads, or other modules in future work.
>
> ## Automatic method for determining the retaining ratio (Q3)
>
> We find this question quite similar to the Question 4. We have not introduced an automatic method for determining the retaining ratio for the clarity of our method. It deserves further exploration in future work.
>
> We hope that the provided analysis and discussion have addressed your concerns.

---

> ### Comment · Reviewer_jzk2 · 2024-11-25
>
> The authors provided helpful clarifications to some of my questions (specifically about the NIAH task), and the proposed method is interesting. However, the results are not particularly strong, and no results are shown for context lengths exceeding 32k or on larger models. For these reasons, I am keeping my rating marginally below acceptance.

---

> > ### Author Response · Authors · 2024-11-30
> > **Performance on Llama3**
> >
> > Dear reviewer,
> >
> > As you are interested in the performance of FreqKV on other models, we have conducted experiments on Llama3-8B-Instruct to validate its effectiveness.
> >
> > The original context window length of Llama3-8B-Instruct is 8K. It is equipped with GQA (Grouped-Query Attention), which means it has a lower proportion of parameters for attention modules than Llama2-7B-Base/Chat (MHA). We use FreqKV to extend the context window from 8K to 16K and evaluate the performance on six Single-Doc QA and Summarization tasks from LongBench. The results of the vanilla model and the SOTA KV compression method SnapKV are also reported in the following table.
> >
> > |  models  |Narrative QA|Qasper|MultiFieldQA-en|GovReport|QMSum|MultiNews|
> > |--|--|--|--|--|--|--|
> > |Llama3-8B-Instruct|**22.52**|31.83|41.04|28.87|**23.25**|26.46|
> > |+ SnapKV|22.27|31.93|41.03|28.77|23.14|26.62|
> > |+ FreqKV|20.55|**32.59**|**45.53**|**30.86**|22.61|**28.19**|
> >
> > Llama3 and SnapKV are evaluated with a context length of 8K. Their performance drops significantly when setting the context length to 16K. FreqKV is evaluated with a context length of 16K. While SnapKV maintains the performance of the vanilla model, FreqKV enables the model to handle longer contexts and achieves improvements on most tasks. It demonstrates the effectiveness of FreqKV when applied to Llama3-8B-Instruct.
> >
> > Moreover, we evaluate FreqKV and SnapKV on **Needle-in-a-Haystack** with the document length ranging from 1K to 16K. FreqKV performs well in extending the context window and achieves an average accuracy of 95.2%, significantly outperforming SnapKV which achieves only 51.5%.
> >
> > **Detailed results are provided in the Appendix. E of the revised version**. These experimental results demonstrate the effectiveness of FreqKV when applied to Llama3-8B-Instruct.
> >
> > We hope that the provided results on Llama3 have addressed your concerns.

---

> ### Author Response · Authors · 2024-11-25
>
> > For these reasons, I am keeping my rating marginally above acceptance.
>
> We appreciate your response and acknowledgement. But we are a little confused about the rating. In ICLR, 6 is marginally above acceptance while your initial rating 5 is marginally below the acceptance (unlike NeurIPS). If you consider our work marginally above acceptance, the rating should be raised.

---

### Official Review · Reviewer_zr3H · 2024-11-04

**Soundness:** 2
**Presentation:** 3
**Contribution:** 3
**Rating:** 5
**Confidence:** 3

**Summary:**

The paper presents FreqKV, a method to extend the context window for large language models (LLMs) by compressing key-value (KV) caches in the frequency domain. The core premise is that the energy distribution of the KV cache concentrates primarily on low-frequency components, allowing high-frequency elements to be discarded without significant information loss. This iterative compression method, applied when the cache reaches a predefined limit, aims to maintain efficient inference without introducing new parameters or modifying the LLM's architecture. The authors claim that FreqKV offers improved memory and computational efficiency in long context tasks.

**Strengths:**

- Novel Approach: The use of frequency domain compression for KV cache management in LLMs is an innovative concept, particularly given the need for extended context handling in generative models.
- Parameter Efficiency: FreqKV avoids adding parameters or modifying the model architecture, making it potentially applicable to existing LLMs without extensive retraining.
- Empirical Validation: Results show comparable perplexity with full KV cache methods, suggesting that FreqKV achieves reasonable performance with reduced memory and computational overhead.
- Benchmark Evaluation: Extensive testing on long context language modeling and understanding tasks provides a solid empirical basis for evaluating FreqKV’s effectiveness.

**Weaknesses:**

- Unclear Decompression Process: The method relies on iterative decompression via the inverse discrete cosine transform (IDCT) to restore KV states for attention computation. This raises concerns about the increased memory and computation needed to reconstitute the full KV tokens, potentially nullifying the benefits of compression.
- Inadequate Justification of Training Requirement: Despite the focus on compression, the paper does not provide clear reasoning for additional training. If FreqKV merely discards high-frequency components, the rationale behind training to learn this transformation remains ambiguous.
- Lack of Discussion on Compression Overhead: The authors overlook a discussion on compression/decompression overheads, which could be significant if IDCT operations occur during inference. The efficiency claims are therefore weakened by the omission of such an analysis.
- Ambiguous Memory Savings: In Figure 3, the reported savings from FreqKV are challenging to interpret, given that the KV cache still requires reconstruction for each attention computation. The lack of explicit comparisons with non-compression methods or details on the computational trade-offs reduces the clarity of the benefits.

**Questions:**

- Does decompression occur on-the-fly during inference? If so, the authors should clarify the associated computational and memory overhead of IDCT operations.
- Why is training required for FreqKV? Given that the method primarily involves discarding high-frequency components, it remains unclear why additional training would be necessary.
- How is memory efficiency maintained when reconstructing the full KV tokens? The need to reconstitute compressed contexts for attention suggests a potential bottleneck that could negate the claimed memory savings.
- What is the impact of different retaining ratios on inference time and accuracy? Further insight into how varying the retaining ratio affects both performance and efficiency would improve the comprehensiveness of the evaluation.

---

> ### Author Response · Authors · 2024-11-23
> **Response to reviewer zr3H**
>
> Dear reviewer, thanks for taking the time to review our paper and for your insightful comments. We will discuss your concerns and refine our paper accordingly.
>
> ## Misunderstanding of our compression process (Weaknesses and Q3)
>
> Most of the weaknesses and questions you mentioned are due to the misunderstanding of our compression process. Our work does not decompress the compressed KV for attention computation with IDCT.
>
> As introduced in Sec. 4.2, KV is compressed in the frequency domain. The whole compression process consists of DCT, filtration of high-frequency components, and IDCT. KV is transferred to the frequency domain by DCT, and then compressed. **IDCT is used to convert the frequency components back to the time domain (Line 202). It does not increase the size of KV which has been compressed.** Therefore, the memory and computation will be saved for the future decoding procedure.
>
> ## Compression overhead (Q1)
>
> Our compression occurs on-the-fly during inference. As illustrated in Sec. 4.3, when the cache is not filled, subsequent tokens could get in and fill the cache. When the cache is filled, it will be compressed so that more tokens could get in until the cache is filled again.
>
> We maintain S = 4 sink tokens uncompressed. The retaining ratio γ in compression is set to 0.5. Therefore, the retaining size during each compression is L = γ · (N − S) = 2046. As long as the cache size reaches its capacity of N = 4096, the 4092 states since the 5-th state in the cache will be compressed into 2046 states. The compression is performed every N − L − S tokens with the complexity of O(NlogN).
>
> To quantify compression overhead, we have measured FLOPs (TFLOPs) with different context lengths. "Sink+Recent" stands for keeping sink tokens and the latest tokens in the cache. It shares the same sink size and retaining size as FreqKV. The difference in FLOPs between the two methods shows the overhead of compression. The statistics are given in the following table. It shows that the computation overhead of our compression process grows less than 0.5% even with a length of 16K and could be negligible.
>
> |   context length   |   4K   |   8K   |   12K   |   16K   |
> |--------------------|--------|--------|---------|---------|
> |      Full KV       | 62.93  | 143.46 |   OOM   |   OOM   |
> |     Sink+Recent    | 62.93  | 125.86 | 188.79  |  251.72 |
> |       FreqKV       | 62.93  | 125.90 | 188.85  |  251.81 |
> |Compression Overhead| 0 (0%) |0.039 (0.031%)|0.064 (0.034%)|0.090 (0.036%)|
> | Compression Times  |    0   |   3    |    5    |   7   |
>
> ## Justification of training requirement (Q2)
>
> As shown in the formula of Eq.1 and 2, DCT and IDCT can be regarded as operations of mixing tokens. We only retain low-frequency components and remove high-frequency components for compression. The elements in the compressed KV are not the initial tokens, but the mixture of tokens. (If we retain all the components in the frequency domain and conduct IDCT, the tokens are the same as the initial tokens.) Moreover, our work aims to extend the context window for a pre-trained LLM. For contexts out of the original window size, the model only uses the compressed KV. Therefore, additional training would be necessary for models to learn to leverage KV states that are compressed in the frequency domain when extending the context window.
>
> We have also measured the performance (PPL) of FreqKV without training on Proof-pile with llama2-7b. The scores are reported in the following table. It shows that the model benefits from the training procedure and learns the new pattern introduced in KV.
>
> | evaluation length |  2048  |  4096  |  8192  |
> |-------------------|--------|--------|--------|
> |      Full FT      |  3.14  |  2.85  |  2.66  |
> |FreqKV (w.o. training)|3.41 |  3.09  |  3.43  |
> |FreqKV (w. training)| 3.16  |  2.88  |  2.80  |
>
> ## Impact of different retaining ratios (Q4)
>
> As given in Sec 6.1, we have conducted detailed studies into the impact of different retaining ratios including computation cost, inference time, and perplexity evaluation. It should be noted that although the FLOPs of different retaining ratios presented in Table 4 are very close, larger retaining ratios lead to smaller chunk sizes, which determines how many attention scores will be masked. As a result, the attention matrix becomes denser and costs an increase in inference overhead with a minimal performance improvement as the retaining ratio increases from 0.5 to 0.75. This study justifies our choice of setting a 0.5 retaining ratio as the default for effectiveness and efficiency.
>
> We hope that the provided analysis and discussion have addressed your concerns.

---

> ### Comment · Reviewer_zr3H · 2024-11-26
> **After response comments**
>
> I appreciate the authors' efforts in preparing the responses. Although the author's responses partially addressed my concerns, I still concur with the other reviewers' comments about the limitation in evaluation, especially in demonstrating the effectiveness of the proposed algorithm in the other models. Therefore, I will maintain my original score.

---

> > ### Author Response · Authors · 2024-11-30
> > **Performance on Llama3**
> >
> > Dear reviewer,
> >
> > Besides responses to your previous questions, we have applied FreqKV to Llama3-8B-Instruct as you are interested in the performance on other models.
> >
> > The original context window length of Llama3-8B-Instruct is 8K. It is equipped with GQA (Grouped-Query Attention), which means it has a lower proportion of parameters for attention modules than Llama2-7B-Base/Chat (MHA). We use FreqKV to extend the context window from 8K to 16K and evaluate the performance on six Single-Doc QA and Summarization tasks from LongBench. The results of the vanilla model and the SOTA KV compression method SnapKV are also reported in the following table.
> >
> > |  models  |Narrative QA|Qasper|MultiFieldQA-en|GovReport|QMSum|MultiNews|
> > |--|--|--|--|--|--|--|
> > |Llama3-8B-Instruct|**22.52**|31.83|41.04|28.87|**23.25**|26.46|
> > |+ SnapKV|22.27|31.93|41.03|28.77|23.14|26.62|
> > |+ FreqKV|20.55|**32.59**|**45.53**|**30.86**|22.61|**28.19**|
> >
> > Llama3 and SnapKV are evaluated with a context length of 8K. Their performance drops significantly when setting the context length to 16K. FreqKV is evaluated with a context length of 16K. While SnapKV maintains the performance of the vanilla model, FreqKV enables the model to handle longer contexts and achieves improvements on most tasks. It demonstrates the effectiveness of FreqKV when applied to Llama3-8B-Instruct.
> >
> > Moreover, we evaluate FreqKV and SnapKV on **Needle-in-a-Haystack** with the document length ranging from 1K to 16K. FreqKV performs well in extending the context window and achieves an average accuracy of 95.2%, significantly outperforming SnapKV which achieves only 51.5%.
> >
> > **Detailed results are provided in the Appendix. E of the revised version**. These experimental results demonstrate the effectiveness of FreqKV when applied to Llama3-8B-Instruct.
> >
> > We hope that the provided results on Llama3 have addressed your concerns.

---

> > ### Author Response · Authors · 2024-12-02
> >
> > Dear Reviewer,
> >
> > Thank you for the time and attention you've dedicated to reviewing our paper. As the discussion stage is coming to an end, please let us know whether we have addressed your concerns or if any further discussions are needed.
> >
> > We look forward to your feedback and thank you once again for your valuable contribution to our work.
> >
> > Best, Authors

---

### Author Response · Authors · 2024-11-25
**Response to all**

Dear Reviewers,

Thank you once again for your valuable comments and suggestions on our paper. We have revised our manuscript and conducted additional experiments based on your feedback. For the reviewers’ convenience, we have highlighted the significant changes in the revised manuscript in SeaGreen. We present a summary of the modifications as follows:

1. **Rusults on Needle-in-a-Haystack (Appendix C)**: We have evaluated FreqKV on the Needle-in-a-Haystack task. FreqKV outperforms the baseline that trains the model with "Local Attention" (or sink+window) to extend the context window.

2. **Compression Overhead (Appendix D)**: We have provided more detailed statistics of the computation overhead of the compression process with different context lengths.

3. **Suggestions from Reviewer npHy**: We have revised our paper accordingly.

If there are any additional issues or points requiring clarification, please feel free to let us know.

---

### Author Response · Authors · 2024-11-28
**Performance on Llama3**

Dear reviewers,

Thank you all for your valuable suggestions. We are working on applying FreqKV to Llama3-8B-Instruct these days as most reviewers are interested in the performance on other models.

The original context window length of Llama3-8B-Instruct is 8K. It is equipped with GQA (Grouped-Query Attention), which means it has a lower proportion of parameters for attention modules than Llama2-7B-Base/Chat (MHA). We use FreqKV to extend the context window from 8K to 16K and evaluate the performance on six Single-Doc QA and Summarization tasks from LongBench. The results of the vanilla model and the SOTA KV compression method SnapKV are also reported in the following table.

|  models  |Narrative QA|Qasper|MultiFieldQA-en|GovReport|QMSum|MultiNews|
|--|--|--|--|--|--|--|
|Llama3-8B-Instruct|**22.52**|31.83|41.04|28.87|**23.25**|26.46|
|+ SnapKV|22.27|31.93|41.03|28.77|23.14|26.62|
|+ FreqKV|20.55|**32.59**|**45.53**|**30.86**|22.61|**28.19**|

Llama3 and SnapKV are evaluated with a context length of 8K. Their performance drops significantly when setting the context length to 16K. FreqKV is evaluated with a context length of 16K. While SnapKV maintains the performance of the vanilla model, FreqKV enables the model to handle longer contexts and achieves improvements on most tasks. It demonstrates the effectiveness of FreqKV when applied to Llama3-8B-Instruct.

Moreover, we evaluate FreqKV and SnapKV on **Needle-in-a-Haystack** with the document length ranging from 1K to 16K. FreqKV performs well in extending the context window and achieves an average accuracy of 95.2%, significantly outperforming SnapKV which achieves only 51.5%.

Detailed results are provided in **the appendix E of the revised version**. These experimental results demonstrate the effectiveness of FreqKV when applied to Llama3-8B-Instruct.

We hope that the provided results on Llama3 have addressed your concerns.

---

### Meta-Review · Area_Chair_HSP4 · 2024-12-20

**Metareview:**

The paper introduces FreqKV, a method designed to extend the context window for large language models (LLMs) by compressing key-value (KV) caches in the frequency domain. The core idea is based on the observation that the energy distribution of the KV cache is primarily concentrated in low-frequency components, enabling high-frequency elements to be discarded without significant loss of information. This iterative compression technique, triggered when the cache reaches a predefined limit, aims to ensure efficient inference without requiring additional parameters or architectural modifications to the LLM. The authors claim that FreqKV improves memory and computational efficiency for tasks involving long contexts.

Overall, the majority of reviewers reached a consensus to reject this work. Key concerns include compression overhead, limited performance improvements, the absence of results for long contexts (≥32k), and some experimental shortcomings. Consequently, the current version does not appear ready for publication. We recommend that the authors address these issues comprehensively, as outlined in the reviewers' feedback, to enhance the work’s overall contribution and robustness.

**Additional Comments On Reviewer Discussion:**

I mainly list the key concerns since different reviewers have different concerns.

1)	compression overhead (reviewer zr3H, jzk2, RuV5)
The authors provide experimental results which but do not convince reviewers

2)	limited performance improvement and even degradation (Reviewer RuV5 ).
The authors provide some explanations, but they are not so convincing.

3)	missing results on long contexts (≥32k) (reviewer jzk2, RuV5,)
The authors do not provide extra experimental results due to their limited GPU resources.

Overall, I agree with the reviewers for most of the concerns.

---

### Decision · Program_Chairs · 2025-01-22

Reject